# MULTI-MODULE GRPO:
# COMPOSING POLICY GRADIENTS AND PROMPT OPTIMIZATION FOR LANGUAGE MODEL PROGRAMS

## ABSTRACT

Group Relative Policy Optimization (GRPO) has proven to be an effective tool for post-training language models (LMs). However, AI systems are increasingly expressed as modular programs that mix together multiple LM calls with distinct prompt templates and other tools, and it is not clear how best to leverage GRPO to improve these systems. We begin to address this challenge by defining MMGRPO, a simple multi-module generalization of GRPO that groups LM calls by module across rollouts and handles variable-length and interrupted trajectories. We find that mmGRPO, composed with automatic prompt optimization, improves accuracy by 11% on average across classification, many-hop search, and privacy-preserving delegation tasks against the post-trained LM—and by 5% against prompt optimization on its own. MMGRPO is released as an open-source learning algorithm for compound AI systems.

## 1 INTRODUCTION

Many modern AI systems and agents are increasingly implemented as modular designs, in which one or more modules are responsible for well-specified subtasks that contribute to a broader objective. A canonical example is "multi-hop" deep research, where the system responds to a question by iteratively using a *query generation* LM module to produce a search query, passing that query to a retriever, and finally feeding all iteratively retrieved passages into a *response generation* LM module to produce the final output. Akin to conventional software, the explicit modularization of such systems makes their behavior controllable, testable, parallelizable, and allows for leveraging the priors of the LM differently for each module.

Group Relative Policy Optimization (GRPO; Shao et al. 2024) has recently emerged as a powerful method for fine-tuning language models (LMs) in the final stages of training. By leveraging relative rewards within groups of "reasoning" rollouts that share the same prompt, GRPO offers a simple alternative to Proximal Policy Optimization (PPO; Schulman et al. 2017), with strong empirical results and far lower engineering overhead. However, GRPO was originally designed for single-stage settings where each rollout consists of a single auto-regressive LM call. It remains unclear how to best extend GRPO to compound systems that involve multiple LM calls whose contexts, control flow, and intermediate states differ across rollouts. In this paper, we ask whether post-training RL algorithms such as GRPO could be applied effectively to such multi-module LM programs. As in standard GRPO, we compare rewards only at the level of full-trajectory rollouts. However, applying GRPO to the multi-module setting introduces challenges that do not arise in single-call systems, as each module has its own isolated context. This is unlike simple LLM agents, like ReAct, where a single chat history captures the full trajectory. Rollouts from the same LM program input can differ in both length and structure due to control-flow variation, early termination, or parsing failures. An example of a LM program invocation can be seen in Figure 1.

In response to these challenges, we implement MMGRPO, a simple and extensible framework for applying GRPO to multi-module setups. The core idea is to relax GRPO's requirement for shared inputs by grouping rollouts at the *module level*, aligning structurally comparable module calls across different trajectories. This approach enables GRPO-style policy gradient updates without requiring

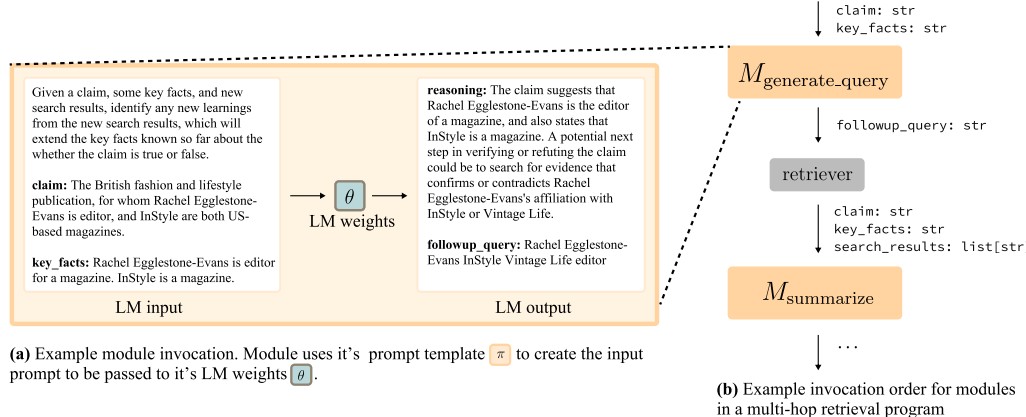

**(a)** Example module invocation. Module uses it's prompt template $\pi$ to create the input prompt to be passed to it's LM weights $\theta$.

**(b)** Example invocation order for modules in a multi-hop retrieval program

Figure 1: Simplified example of a multi-hop retrieval LM program invocation. Each module is responsible for a single subtask, and only its structured outputs (e.g., `followup_query`) are passed to downstream modules. Bold text indicates the specific spans extracted as the module's output. Although each module internally generates tokens to produce these outputs, the raw generation traces (prompts, reasoning steps, and intermediate tokens) remain local to the module and are not shared across the pipeline. An example of a full trace can be found in Section D of the Appendix.

shared histories or module-level inputs across rollouts, and it offers a first strong baseline for online policy-gradient RL methods in this setting. As a result, MMGRPO applies cleanly across a wide range of LM programs and can be used interchangeably with existing multi-module prompt optimization approaches for improving program performance. We open-source MMGRPO as an off-the-shelf optimizer for arbitrary compound AI systems, though we redact those details in this submission.

Because ours is the first implementation of GRPO that applies to sophisticated pipelines of LMs, we are able to conduct a *controlled comparison* of three approaches to optimizing modular AI systems: prompt optimization (PO), online reinforcement learning via MMGRPO, and their combination using the BetterTogether framework (Soylu et al., 2024), which is our main contribution in this work. Our evaluation setup comes from a subset of LangProBe, a Language Programs Benchmark (Tan et al., 2025), and spans three diverse LM program tasks: classification (`Banking77`; Casanueva et al. 2020), multi-hop claim verification (`HoVer`; Jiang et al. 2020), and privacy-conscious delegation (`PAPILLON`; Siyan et al. 2024). Each involves different reasoning styles and control flow structures. Experiments are run using two open-source LMs, `llama3.1-8b-instruct` (Grattafiori et al., 2024) and `qwen3-8b` (Yang et al., 2025).

Across these settings, MMGRPO improves performance by 7% on average against the model's unadapted reasoning performance. While MMGRPO does not always surpass the prompt optimized programs via MIPROv2 (Opsahl-Ong et al., 2024), it complements them effectively: staging MIPROv2 and MMGRPO—à la BetterTogether—consistently yields higher performance than either method alone, improving by 5% and 3% compared to MIPROv2 and MMGRPO, respectively; and by 11% compared to the model's unadapted reasoning performance. These findings suggest that policy gradient RL and PO offer complementary benefits for LM program training, and we advocate for future work exploring their integration in both offline and online settings.

## 2 PRELIMINARIES

GRPO is an online policy gradient method for LM fine-tuning that operates over *groups* of trajectories sharing the *same input prompt*. The GRPO objective encourages the current policy $p_{\theta_{old}}$, parameterized by LM weights $\theta_{old}$, to upweight relatively high-reward completions within a group, while applying PPO-style clipping and KL divergence regularization to ensure stable updates. This results in an updated policy $p_\theta$.

GRPO also makes use of a reference policy $p_{\theta_{\text{ref}}}$ in the KL-divergence penalty, seeking to prevent the updated policy from drifting too far from its original distribution. Here, we express the original GRPO objective in Equation 1 in terms of the prompt–output–reward triples $(q, o_i, r_i)$ to facilitate the extension to the multi-module setting.

$$\mathcal{J}_{\text{GRPO}}(\theta) = \mathbb{E}_{\{(q,o_i,r_i)\}_{i=1}^{G}}$$

$$\frac{1}{G} \sum_{i=1}^{G} \frac{1}{|o_i|} \sum_{t=1}^{|o_i|} \left\{ \min \left( \omega_t \hat{A}_i, \, \text{clip} \left( \omega_t, \, 1 - \epsilon, \, 1 + \epsilon \right) \hat{A}_i \right) - \beta \mathbb{D}_{\text{KL}}[p_\theta \, \| \, p_{\theta_{\text{ref}}}] \right\} \quad (1)$$

where $\theta$ indicates the parameters for an LM shared by all groups

$$\omega_t = \frac{p_\theta(o_{i,t} \mid q, o_{i,<t})}{p_{\theta_{\text{old}}}(o_{i,t} \mid q, o_{i,<t})}, \text{ and } \hat{A}_i \text{ is derived from the observed reward } r_i \text{ (below)}$$

Each GRPO group is defined as a set of triples $\mathcal{G} = \{(q, o_i, r_i)\}_{i=1}^{G}$, constructed by first sampling a fixed prompt from a distribution of questions $q \sim P(Q)$, and then generating a batch of $G$ completions $\{o_i\}_{i=1}^{G} \sim p_{\theta_{\text{old}}}(O \mid q)$ from the current policy. Finally, a scalar reward $r_i$ for each $o_i$ is computed with a reward function. The term $\omega_t$ denotes the importance sampling ratio between the new and old policies for the $t$th token in a given output. The scalar reward $r_i$ is then normalized within the group to compute an advantage $\hat{A}_i$ in the *outcome supervision* formulation of GRPO,

$$\hat{A}_i = \frac{r_i - \text{mean}(\mathcal{R})}{\text{std}(\mathcal{R})}, \quad \mathcal{R} = \{r_i, \text{ reward for } o_i\}_{i=1}^{G}, \quad (2)$$

which is applied uniformly across all tokens $t$ in the corresponding completion $o_i$, as shown in Equation 2.

**LM program formulation** An LM program $\Phi$ is composed of LM modules and other tools orchestrated by its control flow. Let $\mathcal{M} = \{M_1, \ldots, M_{|\mathcal{M}|}\}$ denote the set of LM modules used therein, each of which communicates via natural language.

Given a structured input $x$ (for example, a record with fields such as question or document_titles), executing $\Phi(x)$ orchestrates module invocations, transforming inputs and routing outputs between modules. In other words, $\Phi(x)$ defines a distribution from which we can sample $y, \rho$ pairs, where $y$ is the final output and $\rho$ is the trajectory of module calls:

$$(y, \rho) \sim \Phi(x), \quad \rho = [\zeta_1, \zeta_2, \ldots, \zeta_{|\rho|}] \quad (3)$$

Here, the trajectory $\rho$ records the sequence of module calls, and each trace $\zeta_t = \langle M_t, q_t, o_t \rangle$ captures the module identity as well as the module-level inputs and outputs at module invocation $t$ within the program trajectory. The trajectory $\rho$ logs only the LM-level calls in their execution order and omits any other control logic.

Each module $M \in \mathcal{M}$ is parameterized by a prompt template $\pi_M$ and LM weights $\theta_M$, and may be invoked multiple times. For example, in a multi-hop setting, a module used to generate new queries could be invoked once for each hop, leading to multiple invocations of that module within a single trajectory $\rho$. During execution at module invocation $t$, the prompt template $\pi_{M_t}$ transforms the input $q_t$ into a materialized prompt:

$$q_t \leftarrow \pi_{M_t}(q_t) \quad (4)$$

This prompt is then passed to an LM parameterized by $\theta_{M_t}$, which samples an output $o_t$, returned to the control flow of $\Phi$ for subsequent steps:

$$o_t \sim p_{\theta_{M_t}}(\cdot \mid q_t) \quad (5)$$

This modularity offers several benefits over ReAct-style agents, where every step is appended to a single growing prompt. Such approaches often inflate contexts and leak unnecessary information across steps. Further, they are more difficult to test as the smaller subtasks the agent must execute,

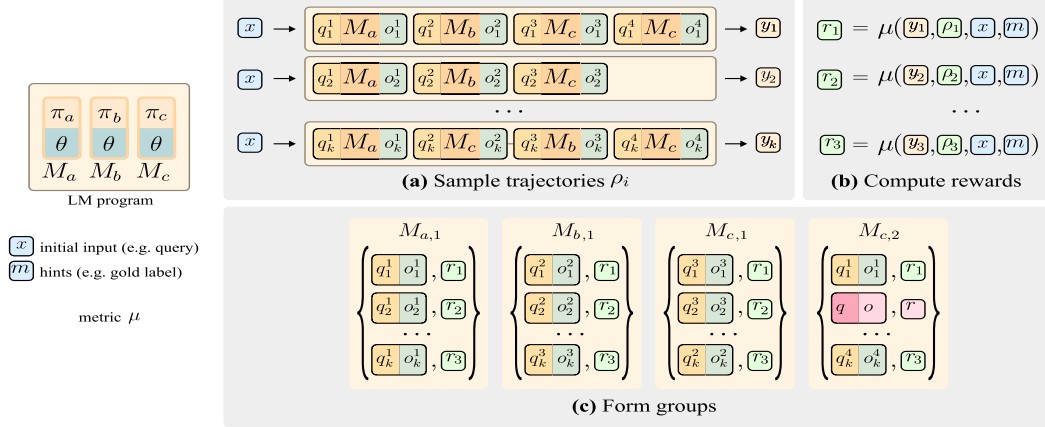

Figure 2: Illustration of MMGRPO's group building strategy. We assume an LM program with modules $M_a$, $M_b$, and $M_c$, an input $x$ (with optional hints $m$), and a final reward metric $\mu$. MMGRPO constructs GRPO groups in three steps: **(a)** Sample $k$ trajectories by running the LM program. For each input, the program logic selects the next module to call and prepares its input $q_1$. After the module returns an output $o_1$, the program logic prepares the next module input, and so on. Because control flow is governed by the program logic, the sequence and number of module calls can differ across trajectories, even for the same $x$. Each run ultimately produces a final output $y$. **(b)** A reward function scores the final output (and optionally the trajectory), producing a scalar reward $r$. **(c)** MMGRPO then forms GRPO groups by collecting all call data for a given module and relative invocation index (e.g., "second call to $M_c$") into a separate group. When a trajectory does not contain a particular module invocation, a padding strategy (e.g. repeating one of the other datapoints in the group with it's associated rewards) is applied so that group sizes remain consistent across trajectories.

such as query generation, cannot be easily evaluated independent of the other subtasks. In contrast, control flow within LM programs specifies exactly which inputs each module sees. This allows repeated module invocations without carrying along unnecessary context, and further allows for each module to be tested in isolation. This explicit modularity makes execution more interpretable and is a core reason why multi-step GRPO formulations, which often assume a single expanding prompt, are not directly applicable to multi-module LM programs (Jin et al. 2025; Zeng et al. 2025; Wang et al. 2025).

**LM program optimization** Let $\mathcal{D} = \{(x, m)\}$ be a dataset of inputs $x$ and optional metadata $m$ (e.g., final answer, documents to retrieve, or PII to redact). The goal is to learn the parameters of the given LM program $\Phi$, namely, the prompt templates $\pi_M$ and LM weights $\theta_M$ for each module $M \in \mathcal{M}$, such that we maximize the expected reward:

$$\mathbb{E}_{(x,m)\sim\mathcal{D};\ (y,\rho)\sim\Phi_{\Pi,\Theta}(x)} \left[ \mu\left(y, \rho, m\right) \right] \tag{6}$$

Here, the reward function $\mu(y, \rho, m)$ scores the execution, typically based on the final output $y$'s correctness. Any metadata $m$ (e.g., gold answers) is not visible to the program during execution but may be used by $\mu$ for evaluation.

## 3 APPLYING GRPO TO MULTI-MODULE LM PROGRAMS

Given a dataset $\mathcal{D}$ and a reward function $\mu$, our goal is to optimize an LM program $\Phi$ consisting of modules $\mathcal{M}$ by updating the weights $\theta_{M_i}$ of each module. In standard GRPO, each group contains trajectories from a single auto-regressive LM call—i.e., one prompt and its full output. However, LM programs typically comprise multiple modules with each one invoking its own LM using a custom prompt. This raises the question of how to best extend GRPO grouping to this multi-module setting. To set a strong baseline in this space, we explore the simplest possible design with MMGRPO, which allows our implementation to remain largely modular with respect to existing GRPO implementations.

MMGRPO starts by sampling full program multiple times, forming a group of program trajectories with each trajectory containing many module invocations. It then aligns module calls across these trajectories and creates GRPO groups at the module level, each containing input–output–reward triples for a specific module. We refer to the group of full program trajectories as the *program-group* to distinguish from the module-level grouping. Adhering to the spirit of the standard GRPO implementation, we uniformly distribute the final reward from a single program trajectory to all modules that were invoked, with the only exception to uniform distribution being when a module failed to adhere to the required structured output such as when an invalid JSON structure was generated or required keys were missing. A modified GRPO loss is then applied independently to each group, updating only the LM weights of the module that produced the group's data. Section 5 shows that this approach is able to improve LM programs on realistic, real-world tasks. We focus on the high-level design in this section, deferring implementation details to Appendix A.

The program-group of trajectories used in MMGRPO consists of multiple executions of the same program on a shared program-level input $x$, i.e., $(y, \rho) \sim \Phi(x)$, where $y$ is the final program output and $\rho = [\zeta_1, \zeta_2, \ldots, \zeta_{|\rho|}]$ is the trajectory of module calls. Each $\zeta_t$ is a triple containing the invoked module $M_t$, the prompt $q_t$ sent to the corresponding module LM $\theta_{M_t}$, and the resulting output $o_t$. The program-level output reward for the entire trajectory is computed as $r = \mu(y, \rho, m)$, where $m$ is any additional metadata associated with the example.

To construct GRPO groups, MMGRPO aligns module calls across trajectories based on both the module identifier and the relative order in which it appears within the trajectory. This alignment process yields module-level GRPO groups, each of the form $\{(q_i, o_i, r_i)\}_{i=1}^{G}$, where $q_i$ and $o_i$ are extracted from a group of aligned traces all generated by a specific module $M$, and $r_i$ is set to the corresponding program-level output reward for the trajectory that generated each trace.

$$\mathcal{J}_{\text{mmGRPO}}(\boxed{\theta_M}) = \mathbb{E}_{\{(\boxed{q_i}, o_i, r_i)\}_{i=1}^{G}}$$

$$\frac{1}{G} \sum_{i=1}^{G} \frac{1}{|o_i|} \sum_{t=1}^{|o_i|} \left\{ \min\left(\omega_t \hat{A}_i, \text{clip}\left(\omega_t, 1-\epsilon, 1+\epsilon\right) \hat{A}_i\right) - \beta \mathbb{D}_{\text{KL}}\left[\boxed{p_{\theta_M}} \| \boxed{p_{\theta_{M_{\text{ref}}}}}\right] \right\} \quad (7)$$

where $\boxed{\theta_M}$ indicates the parameters for an LM shared by all groups

$$\omega_t = \frac{p_{\theta_M}(o_{i,t} \mid \boxed{q_i}, o_{i,<t})}{p_{\theta_{M_{\text{old}}}}(o_{i,t} \mid \boxed{q_i}, o_{i,<t})}, \text{ and } \hat{A}_i \text{ is computed from } r_i \text{ via Equation 2}$$

In practice, not all trajectories generated by $\Phi$ given the same program-level input $x$ follow the same structure; the program logic may diverge (e.g., by invoking different modules or terminating early), or errors such as module-level parsing failures may halt execution. This is exemplified in Figure 2.

To create groups from trajectories with different structures, MMGRPO optionally pads smaller groups to a fixed size before applying the loss, described in more detail in Appendix A. Once the groups are formed, MMGRPO loss in Equation 7 is applied independently to each module-level group, with two key differences from the original GRPO objective (Equation 1). First, rather than updating a shared LM, each group updates only the weights of the module it corresponds to. Second, unlike GRPO where completions share a single prompt, datapoints in a module-level group may have different prompts $q_i$, reflecting variation in upstream context.

**Markovian assumption** Let $\rho_{<i}$ denote the trajectory prefix up to module $i$ for trajectory $\rho$. After the program logic selects the next module $M$ and input $q_i$, we assume that the next-token distribution depends only on the local state of the module:

$$p(o_{i,t} \mid \rho_{<i}, q_i, o_{i,<t}) = p_M(o_{i,t} \mid q_i, o_{i,<t}),$$

so the importance ratio $\omega_t$ can be computed using the local conditionals as in

$$\omega_t = \frac{p_{\theta_M}(o_{i,t} \mid q_i, o_{i,<t})}{p_{\theta_{M_{\text{old}}}}(o_{i,t} \mid q_i, o_{i,<t})}.$$

This relies on treating routing by the program logic as observed (i.e., probability of selecting the module that's actually selected equals to 1, $p_{\text{mo}}(M \mid \rho_{<i}) = 1$, along the executed trajectory). If

program logic was stochastic (e.g. generating a random number and deciding which module to call next using it), additional treatment would be required. We assume the simple case in our work and assume that all stochasticity in the program is contained in the LM generations.

**Module vs. trajectory-level groups** A detailed comparison between trajectory- and module-level formulations for MMGRPO is provided in Appendix B under three assumptions: (1) a single shared LM is trained (one set of weights $\theta_M$); (2) trajectories can be linearized with equal structure (no parallel or asynchronous calls); and (3) the only source of stochasticity is LM generation (program is deterministic given the same generations). Under these conditions, the trajectory-level formulation and the sum of per-module formulations have the same gradient *direction* and differ only by a normalization factor (per-trajectory vs. per-module averaging).

We adopt the module-level formulation for two reasons. First, it is simpler and more composable: factorizing training into per-module GRPO terms that use only local contexts yields a modular implementation that is drop-in compatible with GRPO variants and avoids the cross-module masking required by trajectory-level updates when multiple LMs are present. Second, module-level normalization scales each update by its *own* output length rather than by the total trajectory length, preventing long outputs elsewhere in the pipeline from dominating the gradient. While the trajectory-level objective is the classical policy-gradient estimator, the module-level objective can be viewed as a reweighted estimator that preserves gradient direction under our assumptions (differing only in scale). Moreover, when different modules use different LMs $\{\theta_{M_1}, \theta_{M_2}, \theta_{M_3}\}$, a trajectory-level update for $\theta_{M_1}$ either requires masking tokens not produced by $M_1$ or exposes other modules' text—both undesirable for privacy; whereas the module-level view updates each $\theta_M$ from its own invocations with no cross-module leakage, while still propagating the trajectory-level reward via $\hat{A}_i$.

# 4 Composing Online RL with Prompt Optimization via BetterTogether

BetterTogether (Soylu et al., 2024) demonstrates that combining prompt optimization (PO) with weight optimization yields stronger results than using either technique alone, specifically in the context of offline RL via rejection fine-tuning on outcome-filtered trajectories. Rather than applying weight optimization directly to an unmodified program, the authors first optimize the program's prompt templates and then apply weight optimization on the resulting prompt-optimized program.

We extend this approach to the online RL setting using MMGRPO, and combine it with a state-of-the-art prompt optimizer, MIPROv2 (Opsahl-Ong et al., 2024). While Soylu et al. (2024) also experiment with alternative compositions, such as running prompt optimization after weight tuning, in this work we focus on the former: applying MMGRPO to a prompt-optimized program.

# 5 Experiments

## 5.1 LMs and datasets

For both LM programs and datasets, we start with implementations from the Language Programs Benchmark (LangProBe), which was created to allow easy evaluation of optimization approaches for multi-module LM programs (Tan et al., 2025). The DSPy implementation of these LM programs for each task are shown in Appendix C. For more information on the LMs and datasets used along with their license information, refer to Appendix E.

**LMs** We run our experiments on two open LMs: `llama3.1-8b-instruct` (Grattafiori et al., 2024) and `qwen3-8b` (Yang et al., 2025). Although MMGRPO allows for different LM copies to learn separate weight updates for the different modules of a program, we focus on the multi-task setting where a single set of weights are shared for all modules in a LM program.

**Classification** `Banking77` is an intent classification benchmark involving $13,083$ labeled customer service queries from the banking domain Casanueva et al. (2020). The task is to assign each user query to one of 77 intent classes, example of which is shared below.

```
Example datapoint for Banking77

text: I have been waiting over a week. Is the card still coming?
label: card_arrival
```

We implement a simple program for this task using a single Chain-of-Thought (CoT) module (Wei et al., 2022), which first produces a reasoning trace before predicting the intent label. For evaluation we compute the exact match between the ground-truth label and the generated label. Since the program we have for Banking77 has only a single module, running the MMGRPO algorithm on it is the same as the standard GRPO setup. For training and evaluation, we randomly sample 250 training examples and 500 for development.

**Privacy-conscious delegation**   The Private User Prompt Annotations (PUPA) benchmark constructed by Siyan et al. (2024) focuses on privacy-preserving question answering, where the goal is to respond to user queries without exposing private information to external APIs.

```
Example datapoint for PAPILLON

user_query: Send a friendly e-mail opening to Sara telling her thank you for calling me
back so quickly, taking the time to discuss with me FRC's IPAEP, and for your immediate
willingness to make reasonable accommodations to help break down the barriers caused
by my disabilities so that I can have equal access to your program. Your kindness and
compassion is appreciated.
target_response: Dear Sara, I hope this email finds you well. I just wanted to take a
moment to express my sincere gratitude for your prompt response to my call and the time
you took to discuss FRC's IPAEP with me. I truly appreciate your immediate willingness
to make reasonable accommodations to ensure that I have equal access to your program
and to help break down the barriers caused by my disabilities...
pii_str: sara||ipaep||frc
```

We use PAPILLON, also from Siyan et al. (2024), a two-module pipeline that generates a redacted version of a private user query, sends the redacted query to an untrusted but more performant external model, then uses the response of that external model to generate the final response. We utilize openai/gpt-4.1-mini-2025-04-14 (OpenAI, 2025) as the external LM. As described in Siyan et al. (2024), the evaluation metric is a composite score which takes into account the content of the response and the amount of private information that was leaked, both of which are scored by a large LM acting as a judge. We evaluate this setup using 111 training examples and 221 for development.

**Multi-hop claim verification**   HoVer (Hoppy Verification, Jiang et al., 2020) is a claim verification benchmark where the task is to extract facts from multiple relevant Wikipedia articles and deciding whether a given claim is supported.

```
Example datapoint for HoVer

claim: This director is known for his work on Miss Potter. The Academy of Motion
Picture Arts and Sciences presents the award in which he was nominated for his work
in "Babe".
titles: ['Miss Potter', 'Chris Noonan', 'Academy Award for Best Director']
```

The claims in HoVer are *multi-hop* in that they require multi-hop reasoning by connecting information found in different articles. The original dataset has $18,171$ train and $4000$ development and test examples derived from the examples in the HotPotQA dataset (Yang et al., 2018). Our program for HoVer consists of 2 modules, a query generation module and a fact summarization module, called iteratively over 4 hops, along with a ColBERTv2 (Santhanam et al., 2021) retriever indexed on the short snippets from the Wikipedia (2017) dump provided with the HotPotQA dataset, shared with HoVer. We refer to the particular 4-hop variant Hover program we use with $HoVer_{4-HOP}$, in order to differentiate it from the one provided in Tan et al. (2024). The program returns up to 100 passages at the end, and the final metric evaluates whether the gold passages are found within the returned passages using Recall@100. We build our splits from the original train split, randomly sampling 500

examples each for our train and development splits; while ensuring that we don't sample any two examples derived from the same HotPotQA question.

## 5.2 BASELINE AND METHOD DETAILS

We evaluate each of our LM and task pairs with vanilla Chain-of-Thought (CoT) and a prompt optimizer, to serve as baselines. We demonstrate our MMGRPO optimizer in two flavors: MMGRPO, and BetterTogether MMGRPO. While each method assumes access to a program-level evaluation metric, none relies on an external oracle dataset. Instead, we generate training data dynamically by running the program itself and bootstrapping from model outputs and associated program-level metrics.

We use the DSPy framework (Khattab et al., 2024) to run our baseline experiments and develop our new MMGRPO optimizers. We use DSPy's RL training library, Arbor (Ziems et al., 2025), which draws inspiration from the Verifiers library (Brown, 2025).

**Inference**   We use the vLLM (Kwon et al., 2023) engine for sampling with max context length of $32,768$ tokens for inference. We set max tokens to $1032$ and re-try each query up to 3 times in case of module parsing errors. For qwen3-8b, we use sampling_temperature $= 0.6$, top_p $= 0.95$ and top_k $= 20$ following the parameters used for its instruction training as noted in Yang et al. (2025). For llama3.1-8b-instruct, we use sampling_temperature $= 0.6$ and top_p $= 0.9$ following the official model card's generation configuration in HuggingFace (MetaAI, 2024).

**Vanilla CoT**   We adopt the Chain-of-Thought (CoT) prompting method introduced by Wei et al. (2022), where each module's prompt instructs the language model to first generate a *reasoning* field before producing its final output (e.g. answer, search_query). This creates a strong prompt baseline to compare our methods against. Unless stated otherwise, both the prompt-optimization and MMGRPO methods described below begin training from this base CoT prompt. We refer to this initial prompt configuration as the "Vanilla CoT" program.

**MIPROv2**   We use the state-of-the-art prompt optimizer Multiprompt Instruction PRoposal Optimizer Version 2 (MIPROv2; Opsahl-Ong et al. 2024) as our prompt-optimized baseline. MIPROv2 jointly optimizes module prompts—free-form instructions and optional few-shot examples—by (i) bootstrapping candidate exemplars, (ii) proposing instructions, and (iii) selecting high-yield instruction–example combinations via Bayesian optimization over program-level metrics. For our experiments, we use the auto=medium setting, which uses 12 trials; 12 few-shot and 6 instruction candidates, and automatically uses a 80% of the train set for validation. We refer to the program optimized using MIPROv2 with these settings as the prompt-optimized program and re-use it for the BetterTogether strategy below.

**mmGRPO**   We train our models using the HuggingFace GRPOTrainer, each with a maximum context length of 8192 tokens. Training is performed with a temperature of 0.6, a learning rate of $1 \times 10^{-5}$, gradient accumulation steps of 20, with per device train batch size of 1. We use $\beta = 0.01$ and gradient norm clipping of 0.1 for qwen3-8b; and $\beta = 0.04$ and gradient norm clipping of 0.5 for llama3.1-8b-instruct.

We run MMGRPO for 750 steps, using 4 training examples per step. At each step, we randomly draw 4 examples from the training dataset. For each example, we generate 12 rollouts, which are then grouped into module-level GRPO groups using the procedure in Algorithm 2. We use a train context length of $8,192$ tokens, which is used to filter any trajectory with a module level prompt and completion longer than this. We apply Low-Rank Adaptation (LoRA, Hu et al. 2021) with rank r $= 16$, lora_alpha $= 64$, lora_dropout $= 0.05$, targeting the projection modules $[q, k, v, o, up, down, gate]$. We run all of our MMGRPO experiments below using these same settings. Pseudocode of the MMGRPO algorithm can be found in Algorithm 1.

**mmGRPO with BetterTogether**   We further evaluate a setting that combines prompt optimization with the weight optimization of MMGRPO following the BetterTogether framework (Soylu et al., 2024). Specifically, instead of optimizing the LM program's weights directly, we first apply prompt optimization to identify high quality prompts for each module. The prompts are then kept

Table 1: Performance of different learning algorithms across three LM programs: a single-stage intent-classification program, `Banking77`, two-stage privacy-conscious delegation program, `PAPILLON`, and a four-stage retrieval based question answering program `HoVer4-HOP`. MIPROv2 represents a prompt optimization baseline where prompts for all the stages in a program are jointly optimized, while Vanilla CoT refers to vanilla chain-of-thought prompting. Both MMGRPO and MIPROv2 improve over the untuned baseline, though neither consistently dominates the other. The best overall performance is achieved by the BetterTogether variant of MMGRPO, which first applies prompt optimization using MIPROv2 and then fine-tunes using MMGRPO. We report dev set accuracy for each cell, averaged over 3 seeds, with the dev set being strictly for evaluation and not for optimization.

| Strategy | Banking77 | | PAPILLON | | HoVer4-HOP | | Avg Scores | | |
|---|---|---|---|---|---|---|---|---|---|
| | llama3.1 | qwen3 | llama3.1 | qwen3 | llama3.1 | qwen3 | llama3.1 | qwen3 | All |
| *Baseline Strategies*: | | | | | | | | | |
| Vanilla CoT | 58.4 | 64.6 | 76.2 | 78.3 | 59.5 | 60.6 | 64.7 | 67.8 | 66.3 |
| MIPROv2 (PO) | 59.4 | 65.9 | 83.9 | 78.1 | 63.4 | 69.3 | 68.9 | 71.1 | 70.0 |
| MMGRPO *Strategies*: | | | | | | | | | |
| MMGRPO | **63.7** | 64.9 | 83.9 | **83.3** | 60.2 | 71.0 | 69.3 | 73.1 | 71.2 |
| BetterTogether(PO, MMGRPO) | **63.7** | **69.1** | **86.5** | 81.1 | **68.3** | **71.5** | **72.8** | **73.9** | **73.4** |

fixed, and MMGRPO is used to optimize the program's weights. We refer to this configuration as BetterTogether(PO, MMGRPO), which is our main contribution.

## 5.3 MAIN RESULTS

Our main experimental results are shared in Table 1, evaluated on the dev set and averaged over 3 seeds. The dev set is used exclusively for evaluation and plays no role in optimization.

**MMGRPO and BetterTogether(PO, MMGRPO) consistently improve over their respective baselines.** We can see that the MMGRPO row is consistently higher than the "Vanilla CoT" row, 7% on average. Similarly, BetterTogether(PO, MMGRPO) shows consistent gains over the "MIPROv2 (PO)" row, 5% on average. These show that MMGRPO is effective at finding better policies for the provided program across all LM–task pairs.

We believe these results are consistent with a key observation about online RL methods, such as MMGRPO, where their effectiveness depends heavily on the quality of the initial rollouts. When the base policy is too weak, exploration bottlenecks can limit MMGRPO's ability to discover high-reward trajectories, causing poor gains for complex LM programs such as those used for HoVer. Instead, prompt optimization improves the reliability of early trajectories, providing a more favorable initialization for weight updates and allowing MMGRPO to exploit higher-quality rollouts. This also can help explain why BetterTogether consistently surpasses either method alone.

**PO is competitive with lower computational budgets.** When averaged across all tasks and models, MIPROv2 alone improved upon the Vanilla CoT strategy by 5% compared to MMGRPO's 7% improvement. However, MIPROv2 achieved these results significantly faster while using fewer GPU-hours. On average, our vanilla MMGRPO experiments took 18.7 hours using 2 `H100` GPUs whereas MIPROv2 took only 1.4 hours on average and only required 1 `H100` GPU. These results indicate that PO approaches like MIPROv2 are likely much more feasible for settings which have lower computation budgets.

**BetterTogether(PO, MMGRPO) performs the best in most task pairs.** BetterTogether(PO, MMGRPO) approach improves over the Vanilla CoT by 11%, MIPROv2 by 5%, and vanilla MMGRPO by 3%. This shows the value of high-quality rollouts at the start of MMGRPO training, as performing PO generates stronger rollouts, leading to a more robust training signal early in the training runs.

## 6 RELATED WORK

**Prompt optimization** Much recent work adapts prompts to fit data and tasks, i.e., optimizes the prompts used to invoke an LM rather than its weights. Broadly, approaches include (i) *instruction generation*, in which LMs synthesize candidate instruction templates (Yang et al., 2024; Zhou et al., 2023; Pryzant et al., 2023; Fernando et al., 2024); (ii) *gradient-based* methods that optimize discrete text via differentiation (e.g., soft/continuous prompts) and then project back to tokens (Shin et al., 2020; Wen et al., 2023); and (iii) *RL-based* editors that treat prompt edits as actions and optimize them against trajectory-level rewards (Deng et al., 2022; Zhang et al., 2023; Hao et al., 2023), among many others.

**Weight optimization** Proximal Policy Optimization (PPO) has been widely used for post-training language models with reinforcement learning, particularly when aligning language models with human preferences or feedback (Schulman et al., 2017; Ouyang et al., 2022). In the context of Reinforcement Learning from Human Feedback (RLHF), PPO uses an actor-critic approach where a reward model is trained to approximate human preferences and a policy model is optimized based on predicted rewards from the reward model. However, PPO is computationally intensive, as it requires training and running both the policy model and the reward model during optimization.

Recently, Direct Preference Optimization (DPO) algorithms emerged as a simpler alternative that avoids explicit reward modeling and instead learns from contrastive preference pairs (Rafailov et al., 2023). Similarly, Group Relative Policy Optimization (GRPO) offers an efficient alternative to PPO by avoiding the need for a value model and instead relying on estimated advantages through relative rewards within a group of rollouts (Shao et al., 2024).

**Optimization of LM Programs' Prompts & Weights** Existing work has explored optimizing LM programs with prompt optimizers, including those that focus primarily on rejection sampling (Khattab et al., 2024) and others that extend this to use Bayesian optimization for selecting the instruction-demonstration candidates that are most promising (Opsahl-Ong et al., 2024). Additional work (Soylu et al., 2024) has explored combining weight optimizers with prompt optimizers for additional benefit, but in the context of offline RL. However, adapting some techniques to LM Programs requires making a number of decisions (Section 2) and presents substantial implementation challenges. The present work describes how we generalize online RL via GRPO to LM programs composed of multiple modules.

## 7 CONCLUSION

We introduce MMGRPO, a novel extension of GRPO that enables online weight optimization for multi-module LM programs by propagating final rewards backward across disjoint modules. This design preserves GRPO's practicality while accommodating heterogeneous prompts and partial trajectories, with applications such as privacy-conscious modularization in compound AI pipelines and multi-hop deep-research systems. Our experiments demonstrate that MMGRPO is competitive with strong baselines across tasks and models, validating its effectiveness in navigating the challenging credit assignment problem without requiring intermediate supervision. We further show that combining MMGRPO with state-of-the-art prompt optimization methods via BetterTogether yields the strongest overall performance in the majority of settings, revealing that complementary relationship between weight and prompt optimization holds for online RL methods.

**Limitations** While our experiments demonstrate the promise of multi-module RL formulations, this work has limitations. We use 8-billion parameter language models, which may not reflect how MMGRPO performs with larger models, and we also rely on LoRA for fine-tuning; while efficient, this may limit training performance compared to full-parameter updates. Further, we evaluate only one MMGRPO implementation despite many possible alternative formulations. Finally, while Banking77 is a well-understood classification task, we study it in a limited-feedback setting where models only receive rewards derived from bootstrapped rollouts, not supervised intent labels. While supervised training enables encoder models to perform well on this task, we focus on investigating whether current methods can achieve strong performance from reward signals on the rollouts alone. Our results suggest that this is not yet the case.

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

APPENDIX

# A  MMGRPO ALGORITHM

## A.1  OVERVIEW

The MMGRPO algorithm extends GRPO to the multi-module setting by improving the LM weights of each module in a program through module-level policy gradients. Two core abstractions distinguish MMGRPO in Algorithm 1: (1) the ability to sample trajectories from multiple teacher programs, and (2) the construction of module-level GRPO groups based on relative invocation order. These components are highlighted in the algorithm and explained in more detail in Section A.2 and Section A.3, respectively, while the remaining steps follow standard GRPO procedure and are included for completeness.

---

**Algorithm 1** MMGRPO: GRPO for multi-module LM programs

**Require:**

    **Student program** $\Phi$, with modules $M \in \mathcal{M}$
    **Training set** $\mathcal{D}$
    **Metric** $\mu$
    **Teacher programs** $\mathcal{T}$ (optional), defaults to a list containing only the student program if left empty
    **Data collection hyper-parameters** $\Psi_{\text{data}}$ (optional):
        number of training steps $N_{\text{steps}}$
        batch size $B$
        rollout configuration $K : \mathcal{T} \to \mathbb{N}$, specifying the number of rollouts per example for each teacher
    **Model training hyper-parameters** $\Psi_{\text{train}}$ (optional): learning rate $\eta$, weight decay $\lambda$, and others
    **Shared hyper-parameters** $\Psi_{\text{shared}}$ (optional): group size $G$

1: **function** MMGRPO( $\Phi, \mathcal{D}, \mu, \mathcal{T}, \Psi_{\text{data}}, \Psi_{\text{train}}, \Psi_{\text{shared}}$ )
2:     **for** step = 1 to $N_{\text{steps}}$ **do**
3:         $\mathcal{B} = \text{SAMPLEBATCH}(\mathcal{D}, B)$
4:         **for** $(x, m) \in \mathcal{B}$ **do**
5:             $\mathcal{R} \leftarrow$ SAMPLETEACHERROLLOUTS$(\mathcal{T}, K)$
6:             grpo_groups, $\Theta \leftarrow$ FORMMODULELEVELGROUPS$(\Phi, \mathcal{R}, G, \mu, x, m)$
7:             **for** each group $\mathcal{G} \in$ grpo_groups and corresponding module LM weights $\theta_M \in \Theta$ **do**
8:                 Update $\theta_M$ via the GRPO objective in Equation 7 using hyper-parameters $\Psi_{\text{train}} \cup \Psi_{\text{shared}}$
9:     **return** $\Phi$ with the same prompt-templates but improved LM weights, i.e., $\{\pi_{M_i}, \theta_{M_i}^*\}_{i=1}^{|\mathcal{M}|}$

10:
11: **function** SAMPLETEACHERROLLOUTS( $\mathcal{T}, K, x, m$ )
12:     $\mathcal{R} \leftarrow \emptyset$
13:     **for** each teacher program $\Phi^{(t)} \in \mathcal{T}$ **do**
14:         num_samples $\leftarrow K[\Phi^{(t)}]$
15:         **for** $k = 1$ to num_samples **do**
16:             $(y, \rho) \sim \Phi^{(t)}(x)$
17:             $\mathcal{R} \leftarrow \mathcal{R} \cup \{(y, \rho)\}$
18:     **return** $\mathcal{R}$

    Assume SAMPLEBATCH is provided
    Refer to Algorithm 2 for FORMMODULELEVELGROUPS

---

MMGRPO takes as input a student program $\Phi$, a training dataset $\mathcal{D}$, a reward metric $\mu$, an optional set of teacher programs $\mathcal{T}$, and optional hyper-parameters (Line 1). If unspecified, the set of teacher programs $\mathcal{T}$ defaults to a singleton set containing only the student program. At each training step (Line 2), the algorithm samples a batch $\mathcal{B}$ of examples from the training dataset $\mathcal{D}$ using the configured batch size $B$ (Line 3). For each example $(x, m) \in \mathcal{B}$ (Line 4), the algorithm collects rollouts from the teacher programs via the SAMPLETEACHERROLLOUTS function (Line 5), which returns a set of output-trajectory tuples. These rollouts are passed to FORMMODULELEVELGROUPS from Algorithm 2 (Line 6), which constructs module-level GRPO groups and returns them along with the corresponding references to the module-level LM weights $\theta_M$ to be updated. The algorithm then iterates over each group and its associated LM weights (Line 7), and applies the GRPO loss (as defined in Equation 7) independently to each group (Line 8), using the specified training hyper-

parameters. After $N_{\text{steps}}$ iterations, the algorithm returns the updated student program $\Phi$, preserving its original prompt templates while incorporating improved LM weights (Line 9).

## A.2 SAMPLING WITH TEACHER PROGRAMS

In addition to the student program, MMGRPO accepts a list of optional *teacher programs*, which are used to generate the set of trajectories that populate the runs list. At each GRPO step, rather than sampling all rollouts from the student program alone, MMGRPO samples trajectories from a specified mixture of teacher programs. This list must include the student itself. All teacher programs share the same structural interface, meaning they operate over the same LM program and module-level input/output fields, but may differ in their module-level prompt-templates (e.g., alternative instructions or few-shot examples) or LM weights (e.g., larger LMs). These variations enable the MMGRPO framework to support training that is online but partially off-policy, providing greater flexibility in guiding learning using curated or higher-performing policies.

The SAMPLETEACHERROLLOUTS function samples trajectories from each teacher program in $\mathcal{T}$, using a rollout configuration $K$ that specifies the number of rollouts to generate per teacher. This per-teacher control enables flexible data mixtures across programs. For each rollout, the function extracts the final output $y$ and trajectory $\rho$, and collects the resulting $(y, \rho)$ pairs into the rollout set $\mathcal{R}$ returned for training.[1]

## A.3 FORMING MODULE-LEVEL GROUPS

---

**Algorithm 2** FORMMODULELEVELGROUPS: Create module-level GRPO groups for MMGRPO

**Require:**
    **Student program** $\Phi$, with modules $M \in \mathcal{M}$
    **Rollouts** $\mathcal{R} = \{(y_j, \rho_j)\}_{j=1}^{R}$, sampled outputs along with their trajectories
    **Group size** $G$
    **Metric** $\mu$
    **Input** $x$
    **Input metadata** $m$

1: **function** FORMMODULELEVELGROUPS( $\Phi, \mathcal{R}, G, \mu, x, m$ )
2:     grpo_groups_dict $\leftarrow$ DEFAULTDICT(list)
3:     **for** each $(y, \rho) \in \mathcal{R}$ **do**
4:         $r = \mu(y, \rho, m)$
5:         relative_invocation_orders $\leftarrow$ DEFAULTDICT(LIST)
6:         **for** each trace $\zeta = (M, q, o) \in \rho$ **do**
7:             Append $(q, o, r)$ to grpo_groups$[(M, \text{relative\_invocation\_orders}[M])]$
8:             relative_invocation_orders$[M] \mathrel{+}= 1$
9:     grpo_groups_dict $\leftarrow$ PADGROUPS(grpo_groups)
10:    grpo_groups $\leftarrow$ [SELECTKDIVERSEELEMENTS($\mathcal{G}, G$) $\mid \mathcal{G} \in$ VALUES(grpo_groups_dict)]
11:    $\Theta \leftarrow$ [Get $M$'s weights $\theta_M \mid (M, \text{relative\_invocation\_order}) \in$ KEYS(grpo_groups_dict)]
12:    **return** grpo_groups, $\Theta$

    Assume DEFAULTDICT, KEYS, and VALUES are provided
    Refer to Section A.3 for descriptions of PADGROUPS and SELECTKDIVERSEELEMENTS

---

We now describe how MMGRPO constructs GRPO-style groups at the module level for LM programs. Once the rollouts are sampled , MMGRPO construct *module-level* GRPO groups via the FORMMODULELEVELGROUPS function described in Algorithm 2. Each GRPO group is defined as a list of $G \leq R$ triples $\{(q_i, o_i, r_i)\}_{i=1}^{G}$, where each element consists of a module-level input prompt $q$, the corresponding output $o$, and the final trajectory-level reward $r$. In practice, one can use $G < R$, the number of rollouts, to leave room for post-hoc adjustments to group size (discussed later in this section).

---

[1]When using teacher programs to sample trajectories, the modules $M$ recorded in the traces reflect those of the teacher rather than the student program. In practice, however, MMGRPO ensures that the module keys used to form module-level GRPO groups correspond to the student program's modules for each respective teacher module, since it is required that student and teachers programs share the "same structure".

Given the program $\Phi$, the list of output–trajectory tuples $\mathcal{R}$, and the desired GRPO group size $G$, FORMMODULELEVELGROUPS iterates over each output–trajectory pair in $\mathcal{R}$ (Line 3), computing a corresponding score $r = \mu(y, \rho, m)$ (Line 4). If the corresponding trajectory is incomplete, a fallback reward is assigned (e.g., a formatting error penalty). Following this, it iterates over the traces in each trajectory (Line 6). Each trace contributes a triple $(q, o, r)$ consisting of the module-level input, output, and final trajectory reward. This triple is added to the group corresponding to $(M, k)$, where $k$ is the relative invocation index of $M$ in the trajectory (Line 7), where the relative index is incremented after each occurrence (Line 8). To ensure uniform group sizes despite variability in module invocation counts across trajectories, Lines 9 and 10 apply post-processing steps that adjust each group to have exactly $G$ elements, as detailed later in this section. Finally, Line 11 constructs a list of LM weight references, one corresponding to each group, and both this list and the final GRPO groups are returned (Line 12).

As a result, FORMMODULELEVELGROUPS creates GRPO groups by both the module identity and their relative position within the trajectory with respect to the other calls to the same module. Let $K_{M_i, \rho_j}$ denote the number of times module $M_i$ is invoked in trajectory $\rho_j$ for $(y_j, \rho_j) \in \mathcal{R}$; then the total number of GRPO groups formed across all trajectories is $\sum_i \max_j K_{M_i, \rho_j}$, where $M_i \in \mathcal{M}$ for the given `runs`. Each resulting group is a list of module-level $(q, o, r)$ triples, corresponding to structurally aligned invocations of a given module at a specific position in the trajectory. In contrast to standard GRPO, which produces a single group per set of rollouts in single-stage settings, MMGRPO yields a list of groups, one for each module and relative invocation position. To ensure uniform group sizes and handle variation across trajectories, MMGRPO apply two *post-processing* steps: PADGROUPS and SELECTKDIVERSEELEMENTS, described next.

**Handling variably invoked trajectories with PADGROUPS** If every module $M_i$ in the student program is invoked the same number of times $K_{M_i, *}$ across all trajectories $\rho_j$ where $(y_j, \rho_j) \in \mathcal{R}$, then each constructed GRPO group will contain exactly $R$ triples prior to the call to Line 9 in Algorithm 2. For example, suppose the LM program consists of two modules, $M_1$ and $M_2$, and $R = 3$ trajectories are sampled. If, in every trajectory, the program calls $M_1$ exactly twice and $M_2$ exactly once, then MMGRPO will form three GRPO groups: two for $M_1$ (corresponding to its first and second calls) and one for $M_2$. Each of these groups will contain exactly three triples, one from each trajectory, without requiring any padding or truncation. This scenario arises when all executions yield structurally identical trajectories and none encounter parsing or runtime errors.

However, in practice, these conditions may not hold: some modules may be invoked fewer times due to variation in control flow, while others may terminate early due to parsing failures or other runtime errors. In such cases, certain module, module invocation level GRPO groups may contain fewer than $N$ elements. To address this, MMGRPO applies post-processing strategies to ensure that each group has a uniform size, with a call to the PADGROUPS function, described here.

The behavior of PADGROUPS is controlled by a `padding_mode` hyper-parameter (not explicitly noted in the function call to it in Algorithm 1), which supports two values: `truncate` and `fill`. Under the `truncate` strategy, it discards all GRPO groups for module $M_i$ whose invocation index exceeds $\min_j K_{M_i, \rho_j}$, ensuring that only groups with complete representation across all trajectories are retained. Under the `truncate` strategy, it discards all GRPO groups for a module $M_i$ whose invocation index exceeds $\min_j K_{M_i, \rho_j}$, ensuring that only those invocation positions represented in every trajectory are retained. We use the `fill` setting for the experiments reported in this paper.

**Ensuring diversity in groups with SELECTKDIVERSEELEMENTS** After standardizing group sizes across trajectories, MMGRPO further adjust seach group to ensure it contains exactly $G$ elements, the target GRPO group size. Rather than sampling elements uniformly at random, it invokes the SELECTKDIVERSEELEMENTS function, which selects (or duplicates) elements to form a group of size $G$ while maximizing diversity within the group. This function handles both down-sampling (when the group has more than $G$ elements) and up-sampling (when it has fewer), favoring selections that increase reward variance in the sampled prompt-output pairs. Contemporaneously, Xu et al. (2025) propose a similar variance-based selection strategy, demonstrating that promoting diversity in GRPO groups improves held-out generalization.

## B   TRAJECTORY-LEVEL VS. MODULE-LEVEL MMGRPO

The original GRPO objective, shown below for reference, is defined at the trajectory level, where each trajectory corresponds to a single auto-regressive call. We refer to this setting as the single-module case:

$$\mathcal{J}_{\mathrm{GRPO}}(\theta) = \mathbb{E}_{\{(q, o_i, r_i)\}_{i=1}^G}$$

$$\frac{1}{G} \sum_{i=1}^{G} \frac{1}{|o_i|} \sum_{t=1}^{|o_i|} \left\{ \min \left( \omega_t \hat{A}_i, \, \mathrm{clip} \left( \omega_t, \, 1 - \epsilon, \, 1 + \epsilon \right) \hat{A}_i \right) - \beta \mathbb{D}_{\mathrm{KL}}[p_\theta \, \| \, p_{\theta_{\mathrm{ref}}}] \right\} \quad (8)$$

where   $\theta$ indicates the parameters for an LM shared by all groups

$$\omega_t = \frac{p_\theta(o_{i,t} \mid q, o_{i,<t})}{p_{\theta_{\mathrm{old}}}(o_{i,t} \mid q, o_{i,<t})}, \text{ and } \hat{A}_i \text{ is derived from the observed reward } r_i \text{ (below)}$$

For multi-module systems such as LM programs, a trajectory consists of multiple module invocations, each with its own local context. This differs from multi-turn dialogue settings, where context typically accumulates chronologically across turns. The version of MMGRPO studied in this paper applies GRPO by collecting multi-module trajectory data and dividing it into single-stage groups, similar in spirit to the original GRPO formulation. This raises a natural question: what would a trajectory-level formulation of MMGRPO look like, and how—if at all—would it differ from the module-level variant?

To explore this, we consider applying the original GRPO objective (Equation 8) to groups of full multi-module trajectories, rather than to groups of single-module inputs and outputs. Specifically, instead of using $\{(q, o_i, r_i)\}_{i=1}^G$ as in Equation 8, we use groups of full trajectories $\{(\rho_i)\}_{i=1}^G$. For each trajectory $\rho_i$, we define $o_i$ as the set of all module outputs, with $o_{i,j}$ denoting the output of the $j$-th module invocation within the trajectory, and $o_{i,j,t}$ indexing individual tokens. Inputs $q_{i,j}$ follow analogous indexing.

Under this setup, we obtain the (hypothetical) trajectory-level MMGRPO objective shown below. Note that updates are still applied separately for each LM used in the program. To simplify the analysis, we make the following assumptions: (1) all modules share the same LM backend, so only one set of weights $\theta_M$ is optimized; (2) trajectories can be linearized (no parallel/asynchronous module calls) and have equal structure/length; and (3) the only source of randomness is token generation by the LM conditioned on the full trajectory. We also omit the KL and clipping terms for simplicity. Note that $\hat{A}_i$ is computed from the *trajectory-level* return $r_i$; we do not introduce module-local advantages. The resulting loss is shown in Equation 9, and we adopt a similar notation later for the module-level version for ease of comparison.

$$\mathcal{J}_{\mathrm{mmGRPO_{trajectory}}}(\boxed{\theta_M}) = \mathbb{E}_{\{\rho_i\}_{i=1}^G}$$

$$\frac{1}{G} \sum_{i=1}^{G} \frac{1}{\sum_{j=1}^{|\rho_i|} |o_{i,j}|} \sum_{j=1}^{|\rho_i|} \sum_{t=1}^{|o_{i,j}|} \left\{ \left( \omega_t \hat{A}_i \right) \right\} \text{ where } \quad \omega_t = \frac{p_{\theta_M} \left( o_{i,j,t} \mid \boxed{q_{i,j}}, o_{i,j,<t} \right)}{p_{\theta_{M_{\mathrm{old}}}} \left( o_{i,j,t} \mid \boxed{q_{i,j}}, o_{i,j,<t} \right)} \quad (9)$$

$$\mathcal{J}_{\mathrm{mmGRPO_{module}}}(\boxed{\theta_M}) = \mathbb{E}_{\{(\boxed{q_i}, o_i, r_i)\}_{i=1}^G}$$

$$\frac{1}{G} \sum_{i=1}^{G} \frac{1}{|o_i|} \sum_{t=1}^{|o_i|} \left\{ \left( \omega_t \hat{A}_i \right) \right\} \text{ where } \quad \omega_t = \frac{p_{\theta_M} \left( o_{i,t} \mid \boxed{q_i}, o_{i,<t} \right)}{p_{\theta_{M_{\mathrm{old}}}} \left( o_{i,t} \mid \boxed{q_i}, o_{i,<t} \right)} \quad (10)$$

We now compare Equation 10 and Equation 9. In practice, MMGRPO forms groups of full trajectories and then decomposes them into single-stage (module-level) groups for GRPO updates; one trajectory-group therefore yields multiple single-stage groups. If we accumulate gradients over all single-stage groups derived from a given trajectory-group, the resulting summed module-level objective aggregates updates for every module across all trajectories in the group, yielding: Equation 11. Relative to

the basic module-level loss, this expression differs only by an outer summation that ranges over all modules from all trajectories (with corresponding index changes). A direct comparison to the module-level objective is obtained by fixing $j$ to a particular step (thereby "collapsing" back to a module-level view).

$$\mathcal{J}_{\text{mmGRPO}_{\text{module\_summed}}}\left(\theta_M\right) = \mathbb{E}_{\{\rho_i\}_{i=1}^G}$$

$$\sum_{i=1}^{|\rho_i|} \frac{1}{G} \sum_{j=1}^{G} \frac{1}{|o_{i,j}|} \sum_{t=1}^{|o_{i,j}|} \left\{\left(\omega_t \hat{A}_i\right)\right\} \text{ where } \quad \omega_t = \frac{p_{\theta_M}\left(o_{i,j,t} \mid q_{i,j}, o_{i,j,<t}\right)}{p_{\theta_{M_{\text{old}}}}\left(o_{i,j,t} \mid q_{i,j}, o_{i,j,<t}\right)} \quad (11)$$

**Comparison** How does Equation 11 compare to Equation 9? Under our simplifying assumptions (shared LM backend, linearized trajectories, equal-structure trajectories, and omission of KL/clipping), the only difference is the normalization: the summed module-level objective averages over each module's output length, while the trajectory-level objective averages over the entire trajectory's total output length. Consequently, the two objectives differ only by a scale factor in the loss (and hence gradient magnitude)

## C    TASK DETAILS

The DSPy implementations for the LM programs for `Banking77`, `PAPILLON`, and `HoVer` are presented alongside example datapoints in their respective sections. Code snippets assume respective DSPy imports are available.

### C.1    BANKING77

A DSPy program for `Banking77` is shared in Python Snippet 1.

```
1 LABELS = [ ... ]  # List of all output labels
2 Banking77 = dspy.ChainOfThought(f"text -> label: Literal{LABELS}")
```

**Python Snippet 1**: DSPy program for Banking77.

### C.2    PAPILLON

A DSPy program for `PAPILLON` is shared in Python Snippet 2.

```
1 class CraftRedactedRequest(dspy.Signature):
2     """
3     Given a private user query, create a privacy-preserving
           request for a powerful external LLM.
4     The LLM may assist without learning private information about
           the user.
5     """
6
7     user_query = dspy.InputField()
8     llm_request = dspy.OutputField()
9
10
11 class RespondToQuery(dspy.Signature):
12     """
13     Respond to a user query.
14     For inspiration, we found a potentially related request to a
           powerful external LLM and its response.
15     """
16
17     related_llm_request = dspy.InputField()
18     related_llm_response = dspy.InputField(desc="information from
           a powerful LLM responding to a related request")
```

```
19      user_query = dspy.InputField(desc="the user's request you need
            to fulfill")
20      response = dspy.OutputField(desc="your final response to the
            user's request")
21
22
23  class PAPILLON(dspy.Module):
24      def __init__(self, untrusted_model):
25          self.craft_redacted_request = dspy.ChainOfThought(
                CraftRedactedRequest)
26          self.respond_to_query = dspy.Predict(RespondToQuery)
27          self.untrusted_model = untrusted_model
28
29      def forward(self, user_query):
30          llm_request = self.craft_redacted_request(user_query=
                user_query).llm_request
31          llm_response = self.untrusted_model(llm_request)[0]
32          response = self.respond_to_query(
33              related_llm_request=llm_request, related_llm_response=
                    llm_response, user_query=user_query
34          ).response
35
36          return dspy.Prediction(llm_request=llm_request,
                llm_response=llm_response, response=response)
```

**Python Snippet 2**: DSPy program for Papillon.

## C.3 HoVer

A DSPy program for HoVer is shared in Python Snippet 3.

```
1  # Assume that a function called deduplicate is defined
2
3  class GenerateThreeQueries(dspy.Signature):
4      """
5      Given a claim and some key facts, generate up to 3 followup
            search query to find the next most essential clue towards
            verifying or refuting the claim. If you think fewer
            queries are sufficient, generate None for the search query
             outputs you don't need. The goal ultimately is to find
            all documents implicated by the claim.
6      """
7      claim = dspy.InputField()
8      key_facts = dspy.InputField()
9      search_query1 = dspy.OutputField()
10     search_query2 = dspy.OutputField()
11     search_query3 = dspy.OutputField()
12
13
14  class AppendNotes(dspy.Signature):
15      """
16      Given a claim, some key facts, and new search results,
            identify any new learnings from the new search results,
            which will extend the key facts known so far about the
            whether the claim is true or false. The goal is to
            ultimately collect all facts that would help us find all
            documents implicated by the claim.
17      """
18      claim = dspy.InputField()
19      key_facts = dspy.InputField()
20      new_search_results = dspy.InputField()
21      new_key_facts = dspy.OutputField()
```

```
22
23
24  class Hover(dspy.Module):
25      def __init__(
26              self,
27              num_hops=4,
28              k_per_search_query=10,
29              k_per_search_query_last_hop=30,
30              num_total_passages=100,
31          ):
32          # Value is fixed to simplify signature construction in
                presented snippet
33          self.num_search_queries_per_hop = 3
34
35          self.num_hops = num_hops
36          self.k_per_search_query = k_per_search_query
37          self.k_per_search_query_last_hop =
                k_per_search_query_last_hop
38          self.num_total_passages = num_total_passages
39
40          self.rm = dspy.ColBERTv2()
41          self.generate_query = dspy.ChainOfThought(
                GenerateThreeQueries)
42          self.append_notes = dspy.ChainOfThought(AppendNotes)
43
44      def forward(self, claim: str) -> list[str]:
45          key_facts = []
46          committed_docs = []
47
48          for hop_ind in range(self.num_hops):
49              is_last_hop = hop_ind == self.num_hops - 1
50              is_first_hop = hop_ind == 0
51              hop_k = self.k_per_search_query_last_hop if
                    is_last_hop else self.k_per_search_query
52              num_docs_to_keep = (self.num_total_passages - len(
                    committed_docs)) if is_last_hop else self.
                    k_per_search_query
53
54              if is_first_hop:
55                  search_queries = [claim]
56              else:
57                  pred = self.generate_query(claim=claim, key_facts=
                        key_facts)
58                  search_queries = [pred.search_query1, pred.
                        search_query2, pred.search_query3]
59              search_queries = deduplicate(search_queries)
60
61              search_results = [r for q in search_queries for r in
                    search_raw(q, k=hop_k, rm=self.rm)]
62              search_results = sorted(search_results, key=lambda r:
                    r["score"], reverse=True)
63
64              unique_docs = []
65              for result in search_results:
66                  if result["long_text"] not in unique_docs:
67                      unique_docs.append(result["long_text"])
68              unique_docs = unique_docs[:num_docs_to_keep]
69              committed_docs.extend(unique_docs)
70
71              if not is_last_hop:
72                  pred = self.append_notes(claim=claim, key_facts=
                        key_facts, new_search_results=unique_docs)
73                  key_facts.append(pred.new_key_facts)
74
```

```
75              return dspy.Prediction(key_facts=key_facts, retrieved_docs
                    =committed_docs)
```

**Python Snippet 3**: DSPy program for HoVer.

## D  EXAMPLE DSPY TRACE DEMONSTRATING DISJOINT MODULE CONTEXTS

The following illustrates how DSPy executes a multi-module program where each module receives its own locally constructed prompt and thus operates over a disjoint context. Note that for brevity we are showing a full trace from a simplified version of the HoVer program shown in Section C of the Appendix.

**Hop 1 (Key-Fact Extraction Module): System Prompt**

```
Your input fields are:

1. `claim` (str):
2. `key_facts` (str):
3. `new_search_results` (str):

Your output fields are:
4. `reasoning` (str):
5. `new_key_facts` (str):

All interactions will follow this structure:

[[ ## claim ## ]]
{claim}

[[ ## key_facts ## ]]
{key_facts}

[[ ## new_search_results ## ]]
{new_search_results}

[[ ## reasoning ## ]]
{reasoning}

[[ ## new_key_facts ## ]]
{new_key_facts}

[[ ## completed ## ]]

Your objective:
Given a claim, some key facts, and new search results, identify any new
learnings required to verify or refute the claim.
```

**Hop 1 (Key-Fact Extraction Module): User Prompt**

```
[[ ## claim ## ]]
Bruce Geller died in 1978. He developed the American television detective
show that Gail Fisher plays the role of Peggy Fair in.

[[ ## key_facts ## ]]
N/A

[[ ## new_search_results ## ]]
[1] "Gail Fisher ... played Peggy Fair on the detective series Mannix ..."
[2] "Mannix ... developed by executive producer Bruce Geller ..."
[3] "Bruce Bernard Geller (1930-1978) was an American producer ..."
[4] "List of Mannix episodes ... developed by Bruce Geller ..."
```

```
Respond with [[ ## reasoning ## ]],
[[ ## new_key_facts ## ]], and [[ ## completed ## ]].
```

**Hop 1 (Key-Fact Extraction Module): Assistant Output**

```
[[ ## reasoning ## ]]
The search results confirm Bruce Geller's death date and his role in
developing Mannix. They also confirm Gail Fisher's role as Peggy Fair.
These facts clarify the relationships relevant to the claim.

[[ ## new_key_facts ## ]]
1. Bruce Geller (1930-1978) developed the detective show Mannix.
2. Gail Fisher (1935-2000) played Peggy Fair on Mannix (1968-1975).

[[ ## completed ## ]]
```

**Hop 2 (Follow-up Search Module): System Prompt**

```
Your input fields are:

1. `claim` (str):
2. `key_facts` (str):

Your output fields are:
3. `reasoning` (str):
4. `followup_search_query` (str):

All interactions follow this structure:

[[ ## claim ## ]]
{claim}

[[ ## key_facts ## ]]
{key_facts}

[[ ## reasoning ## ]]
{reasoning}

[[ ## followup_search_query ## ]]
{followup_search_query}

[[ ## completed ## ]]

Your objective:
Given a claim and extracted key facts, generate a follow-up search query to
find the next most essential clue for verification.
```

**Hop 2 (Follow-up Search Module): User Prompt**

```
[[ ## claim ## ]]
Bruce Geller died in 1978. He developed the American television detective
show that Gail Fisher plays the role of Peggy Fair in.

[[ ## key_facts ## ]]
<<<
1. Bruce Geller (1930-1978) developed Mannix.
2. Gail Fisher (1935-2000) played Peggy Fair on Mannix.
>>>

Respond with [[ ## reasoning ## ]],
[[ ## followup_search_query ## ]], and [[ ## completed ## ]].
```

**Hop 2 (Follow-up Search Module): Assistant Output**

```
[[ ## reasoning ## ]]
The key facts give the dates and roles of Geller and Fisher. The next step
is to verify the exact nature of Geller's contribution to developing Mannix.

[[ ## followup_search_query ## ]]
"Bruce Geller developed Mannix" "Mannix Gail Fisher Peggy Fair"

[[ ## completed ## ]]
```

## E  ASSET INFORMATION

The license information for the models and datasets we used are shared below. All models and datasets are access via HuggingFace.

**qwen3-8b** is shared with the `Apache License 2.0`, accessed via the HuggingFace model identifier `Qwen/Qwen3-8B`

**llama3.1-8b-instruct** is shared with the `Meta Llama 3 Community License`, accessed via the HuggingFace model identifier `meta-llama/Meta-Llama-3.1-8B-Instruct`

**Banking77** is shared with `CC BY 4.0` license

**HoVer** is shared with `CC BY 4.0` license

**PAPILLON** is shared with the `MIT License` license

## F  LLM USE DISCLOSURE

We used Large Language Models (LLMs) in two ways: (i) to polish author-written prose (grammar, phrasing, and concision), and (ii) to brainstorm variations of design and implementation choices and to draft clearer explanations of these choices. All experiments, and results are author-generated. All LLM-suggested text was reviewed, verified, and, rewritten by the authors.

