# OpenReview forum: "Multi-module GRPO: Composing Policy Gradients and Prompt Optimization for Language Model Programs"
_ICLR.cc/2026/Conference — Submitted to ICLR 2026_

### Official Review · Reviewer_eeBe · 2025-10-21

**Soundness:** 2
**Presentation:** 3
**Contribution:** 2
**Rating:** 6
**Confidence:** 4

**Summary:**

This paper introduces MMGRPO, a practical extension of Group Relative Policy Optimization (GRPO) designed for post-training multi-module language model programs. The core contribution is a module-level grouping strategy that enables policy gradient updates across structurally diverse trajectories, and the authors demonstrate that combining this method with prompt optimization yields strong performance improvements on a variety of tasks.

**Strengths:**

- The paper addresses the timely and relevant problem of optimizing compound AI systems, which is a clear departure from traditional single-call LM fine-tuning.
- The proposed MMGRPO method is an intuitive and elegant extension of the existing GRPO framework, making it straightforward to understand and implement.
- The empirical evaluation is solid, demonstrating the method's effectiveness across three distinct tasks and showing the complementary benefits of combining weight optimization with prompt optimization through the BetterTogether framework.

**Weaknesses:**

- My main concerns lie in the uniform credit assignment strategy. Assigning the final trajectory's reward to every intermediate module call is a considerable *simplification*. In multi-step reasoning or tool-use scenarios, the importance of each module's contribution can vary greatly; an early-stage error might derail the entire process, while a late-stage refinement might only offer marginal improvement. This approach masks the complexity of temporal credit assignment and may not be robust for tasks with longer horizons or more intricate module dependencies, where the signal-to-noise ratio for early modules would be low.
- The method's reliance on a deterministic program structure limits its applicability to more dynamic agentic systems where the control flow itself might be learned or stochastic.
- The experiments are conducted on 8-billion parameter models, and it is unclear how these findings on policy optimization will translate to much larger foundational models.

**Questions:**

- Instead of grouping module calls by their relative invocation order, have the authors considered alternative alignment strategies, such as grouping based on the *semantic similarity* of module inputs, which might create more coherent groups for policy learning?
- Given the challenge of sparse rewards in long-horizon tasks, was any form of *reward shaping* or intermediate reward mechanism explored?
- How does the framework differentiate between a low-reward trajectory resulting from a single catastrophic failure versus one from an accumulation of minor imperfections?

---

> ### Author Response · Authors · 2025-12-03
>
> We thank the reviewer eeBe for taking the time to review this work. It is encouraging that the reviewer highlights the problem of optimizing compound AI systems as timely and different from traditional single-call LM fine-tuning, finds mmGRPO to be an intuitive and elegant extension of GRPO that is straightforward to implement, and views the empirical evaluation as solid as we demonstrate effectiveness across three tasks and the complementary benefits of combining weight optimization with prompt optimization in the BetterTogether framework. The remaining concerns are addressed below.
>
> > My main concerns lie in the uniform credit assignment strategy. Assigning the final trajectory's reward to every intermediate module call is a considerable simplification. In multi-step reasoning or tool-use scenarios, the importance of each module's contribution can vary greatly; an early-stage error might derail the entire process, while a late-stage refinement might only offer marginal improvement. This approach masks the complexity of temporal credit assignment and may not be robust for tasks with longer horizons or more intricate module dependencies, where the signal-to-noise ratio for early modules would be low.
>
> Our choice to use only an outcome-level reward and to avoid explicit credit assignment is motivated by consistency with the original GRPO formulation, where an outcome reward is propagated to all tokens in a sampled completion. The same conceptual issue also exists in this context: some tokens contribute more to the error than others, yet they all receive the same outcome level reward. In practice, for many compound AI systems, intermediate labels are not available, and requiring them would significantly limit the applicability of the method.
> Nonetheless, we agree with the reviewer that it is possible that a more elaborate credit assignment scheme would in fact lead to stronger results. That, however, does not conflict with any claims we are making in this work and would make for an excellent future extension. (We do not claim that we have designed the strongest possible RL implementation for compound AI systems, only that we are testing the first variant of GRPO in this setting.)
> That said, the module-level mmGRPO formulation we share in our paper and pick as our default is not restricted to outcome level rewards: the same grouping mechanism can be trivially extended to allow for module specific rewards, if available. This would allow for module-specific rewards to be incorporated if they are available.
> Our key contribution is showing that combining  prompt optimization with weight optimization continues to show consistent gains even when the weight optimization is performed with an online RL method, in the context of compound AI systems. We demonstrate that prompt optimization (via the MIPRO prompt optimizer) and GRPO (as implemented with our mmGRPO extension) provide complementary benefits, and combined together, yield stronger results. We clarify that we describe mmGRPO as a necessary intermediate step towards extending policy-based online RL methods to multi-module compound AI systems (and it is the first such system).
>
> Table 1 demonstrates three distinct mechanisms by which the BetterTogether strategy yields superior performance:
> 1. Overcoming Exploration Bottlenecks of the base policy: On complex multi-hop tasks, mmGRPO alone struggles if the initial policy is too weak to generate successful trajectories (HoVer: 60.2% vs. Baseline 59.5%). MIPRO effectively provides a "warm start" (63.4%), placing the model in a trainable region. The combination then jumps to 68.3%, showing that mmGRPO can exploit the better starting point to achieve significantly better results than either prompt optimization or reinforcement learning alone.
> 2. Orthogonality of Gains: In the privacy task, both methods achieve an individual score of 83.9%. However, their combination rises to 86.5%. This implies the methods learn complementary strategies solving different (sub)problems.
> 3. Performance Ceiling: While prompt optimization offers a strong baseline (Avg 70.0%), it consistently hits a performance ceiling when the model has not been updated parametrically. Adding mmGRPO reliably unlocks further gains (BetterTogether Avg 73.4%), confirming that weight updates can solve fine-grained optimization problems that are strictly inaccessible to prompt engineering

---

> > ### Author Response · Authors · 2025-12-03
> >
> > > The method's reliance on a deterministic program structure limits its applicability to more dynamic agentic systems where the control flow itself might be learned or stochastic.
> >
> > mmGRPO can work with any LM program, which includes both task-specific programs written by domain experts as well as agentic systems. In particular, we focus on optimizing LM programs, i.e., compound AI systems that make multiple calls to LMs as part of a pipeline rather than a single auto-regressive call. Concretely, LM programs are composed of LM modules, each tackling a sub-task (e.g., search query generation, preparing the final answer, redacting a user query). This modularity lets domain experts decide which aspects are handled declaratively via prompts versus imperatively via program logic. Under this definition, an “agentic” system such as a ReAct scaffold is simply an LM program with a single module (i.e. outputting an observation, a reasoning and an action), called in a loop with program logic implementing the tool calls, so this problem specification does not rule out free-form agentic behavior.
> >
> > > The experiments are conducted on 8-billion parameter models, and it is unclear how these findings on policy optimization will translate to much larger foundational models.
> >
> > The current experiments are restricted to 8B parameter models due to compute constraints.
> >
> > > Instead of grouping module calls by their relative invocation order, have the authors considered alternative alignment strategies, such as grouping based on the semantic similarity of module inputs, which might create more coherent groups for policy learning?
> > > Given the challenge of sparse rewards in long-horizon tasks, was any form of reward shaping or intermediate reward mechanism explored?
> > > How does the framework differentiate between a low-reward trajectory resulting from a single catastrophic failure versus one from an accumulation of minor imperfections?
> >
> > We did not implement this variant in the current paper to keep the core contribution focused on “combining prompt optimization with weight optimization with an online RL method, in the context of compound AI systems” rather than optimizing the grouping strategy itself. However, we agree that grouping by (module, call_order) vs. (module, semantic_similarity) could be an interesting direction for future work: it would directly quantify how much within group context helps.
> > No additional reward shaping or intermediate rewards were used in this work. All three tasks are trained and evaluated with outcome-level rewards, in line with the original GRPO formulation.
> > With the current uniform outcome-level credit assignment, mmGRPO does not explicitly differentiate these two failure modes: both would get low final rewards.
> >
> > We appreciate the time and effort you have put in to review our paper and hope that our response clarifies any further questions you may have.

---

### Official Review · Reviewer_LPF6 · 2025-10-27

**Soundness:** 2
**Presentation:** 3
**Contribution:** 3
**Rating:** 4
**Confidence:** 4

**Summary:**

This paper presents an extension of Group Relative Policy Optimization (GRPO) to a setting where multiple module LM programs are invoked, each with their own prompts/contexts. This is challenging for regular GRPO because of varying number of steps, differing control flows etc. The key idea is to generate full program trajectories, align module calls across trajectories and apply GRPO to each module separately. Results are shown on three tasks - intent classification (Banking 77), privacy conscious delegation (PAPILLON), and multi-hop claim verification (HoVer). Two baseline approaches are compared: a chain of thought (CoT) baseline and prompt optimization (MIPROv2). The proposed method  improves upon CoT and sometimes performs better than MIPROv2. When used in combination with prompt optimization using the better together framework, it consistently outperforms both baselines.

**Strengths:**

* Proposes a novel framework to extend GRPO to a setting where multiple module LMs are called.
* Demonstrates that the method performs better than a chain of thought baseline.
* Shows that the method in combination with prompt optimization performs better than using prompt optimization alone.

**Weaknesses:**

* The method does not outperform the prompt optimization baseline on its own.
* It would be good to report a baseline where GRPO can be applied to the multi-module LM setting in some adhoc manner for a specific setting, e.g. by requiring that rollouts have the same number of steps. Even if such a setting is artificial, it would allow us to measure the benefit of the proposed approach which does not impose such requirements.
* Though the approach is general and allows for multiple teachers, it seems like all evaluations are reported with a single teacher.
* L227: The Markovian assumption is a limitation but is not mentioned in the Limitations section.
* L285: Though the approach allows for different LMs for each module, results are presented only for the scenario where the same LM is used for each module.

**Questions:**

* Sec 5.1: Do any of these setups use more than 1 teacher? i.e. does the list of teacher programs in L666 consist of any teacher other than the student program?
* Sec 5.1 : It would be good to present examples of rollouts for each task, showing the trajectory of module calls, and some examples where these might lead to differing lengths.
* Appendix A3: It would be good to present some examples (inputs/outputs) showing how Algorithm 2 (FormingModuleLevelGroups) works since that is a critical component of the proposed approach.

---

> ### Author Response · Authors · 2025-12-03
>
> We thank the reviewer LPF6 for taking the time to review this work. It is encouraging that the reviewer recognizes the framework as an extension of GRPO to multi-module LM programs, notes that the method improves over a chain-of-thought baseline, and highlights that our proposed approach combining prompt optimization with online RL outperforms prompt optimization alone. The remaining concerns are addressed below.
>
> > The method does not outperform the prompt optimization baseline on its own.
>
> Our key contribution is showing that combining  prompt optimization with weight optimization continues to show consistent gains even when the weight optimization is performed with an online RL method, in the context of compound AI systems. We demonstrate that prompt optimization (via the MIPRO prompt optimizer) and GRPO (as implemented with our mmGRPO extension) provide complementary benefits, and combined together, yield stronger results. We clarify that we describe mmGRPO as a necessary intermediate step towards extending policy-based online RL methods to multi-module compound AI systems (and it is the first such system).
>
> More specifically, Table 1 demonstrates three distinct mechanisms by which the BetterTogether strategy yields superior performance:
> 1. Overcoming Exploration Bottlenecks of the base policy: On complex multi-hop tasks, mmGRPO alone struggles if the initial policy is too weak to generate successful trajectories (HoVer: 60.2% vs. Baseline 59.5%). MIPRO effectively provides a "warm start" (63.4%), placing the model in a trainable region. The combination then jumps to 68.3%, showing that mmGRPO can exploit the better starting point to achieve significantly better results than either prompt optimization or reinforcement learning alone.
> 2. Orthogonality of Gains: In the privacy task, both methods achieve an individual score of 83.9%. However, their combination rises to 86.5%. This implies the methods learn complementary strategies solving different (sub)problems.
> 3. Performance Ceiling: While prompt optimization offers a strong baseline (Avg 70.0%), it consistently hits a performance ceiling when the model has not been updated parametrically. Adding mmGRPO reliably unlocks further gains (BetterTogether Avg 73.4%), confirming that weight updates can solve fine-grained optimization problems that are strictly inaccessible to prompt engineering
>
> > It would be good to report a baseline where GRPO can be applied to the multi-module LM setting in some adhoc manner for a specific setting
>
> GRPO assumes that all trajectories in a group share the same input prompt. In a compound AI system, even if we force rollouts to have the same number of steps, the input to modules (e.g., module 2) will differ across trajectories because it would depend on earlier module outputs and decisions made by the program logic. A “multi-module GRPO” constructed by simply fixing the number of steps would violate GRPO’s core assumption (grouping trajectories with different inputs).
> mmGRPO is, to our knowledge, the first online RL implementation that is  applicable to and has been tested for compound AI systems of even mild complexity (e.g., with just two modules). We also note that in the simplified single-module setting of Banking77, mmGRPO groups would be the same as the groups formed by GRPO; in this case, our BetterTogether approach still provides gains, which isolates the benefit of our method even when a standard GRPO baseline is directly applicable.
>
> > Though the approach is general and allows for multiple teachers, it seems like all evaluations are reported with a single teacher.
> > L285: Though the approach allows for different LMs for each module, results are presented only for the scenario where the same LM is used for each module.
>
> We consider the fact that our approach simply extends to multiple teachers to be a plus, following the prior work in compound system optimization. Similarly, our GRPO formulation does allow multiple training backends to be active at the same time. However, these aren’t the focus of our paper, so we didn’t empirically test their utility.
>
> > L227: The Markovian assumption is a limitation but is not mentioned in the Limitations section.
> > Appendix A3: It would be good to present some examples (inputs/outputs) showing how Algorithm 2 (FormingModuleLevelGroups) works since that is a critical component of the proposed approach.
>
> The writer of the LM program decides which inputs are passed to each module. For example, the programmer could choose to implement an LM program with an agentic scaffold where the full history is passed to each module, where there would be no loss of the prior context.
> We have included a new figure in our main text, Figure 2, that highlights how the module level group forming strategy works.
>
> We appreciate the time and effort you have put in to review our paper and hope that our response clarifies any further questions you may have.

---

### Official Review · Reviewer_5Lbj · 2025-10-30

**Soundness:** 2
**Presentation:** 2
**Contribution:** 2
**Rating:** 2
**Confidence:** 3

**Summary:**

The paper studies reinforcement learning for multi-module LM programs by moving GRPO updates from full trajectories to module-level groups, and articulates when this decomposition is preferable. Trajectory-level GRPO is feasible for fixed, single-shape pipelines, but becomes awkward when control flow varies (skips, repeats, early stops) or when module boundaries are administered separately. The proposed MMGRPO samples full runs, then groups tokens by (module id, k-th invocation) and applies a GRPO loss to that module’s tokens using the final program reward. The method rests on a Markov-given-message assumption: once routed to module M, next-token probabilities depend only on M’s input message q (which is expected to summarize upstream state) and its own prefix; routing and tools are treated as environment. Under three conditions, (one shared LM being optimized, equal or linearizable structure, and determinism given the same generations), the paper shows the sum of per-module gradients has the same direction as the trajectory-level gradient up to scale, which justifies the decomposition in fixed-shape settings. In practice, full-run sampling lets each run contribute to every module that fired and allows reuse of teacher traces by aligning their module calls; with LoRA updates and an optional prompt-optimization stage, results are comparable to trajectory-level training on fixed structures and remain usable when control flow varies, subject to how well q captures the necessary context.

**Strengths:**

1. It's relative easy, practical implementation. Sampling full runs does not require architectural changes; Integrates with LoRA, optional prompt optimization, and teacher traces.
2. Though many assumptions, they are stated properly. The main result is scoped correctly: directional equivalence of trajectory- vs module-level gradients under explicit conditions.

**Weaknesses:**

1. Narrow utility window. Benefits seem tied to variable control flow or administrative boundaries; outside that, a single trajectory loss is simpler and likely stronger.
2. Arbitrary partitioning risk. Module boundaries are designer choices; outcomes can depend on where you cut. The paper offers no merge/split robustness or guidance.
3. Strong Markov-given-message assumption. It is assumed, not validated, that q fully summarizes upstream state; if serialization is lossy, gradients are biased.
4. Heterogeneous groups. Grouping by the k-th call mixes unlike contexts/prompts; apples-to-oranges grouping can inflate variance and misdirect updates.

**Questions:**

1. merge/split: what's the design criteria for choosing the boundary? also include ablation of merge/split of modules.
2. specify the message q scheme. what information about all prefix goes into the message that by right is sufficient to encode everything?
3. following (2), curious to see a state-Incorporated module (learned latent / prompt embedding). Try a learned state appended to q: a small encoder s = f(upstream traces) whose vector is serialized (text or special tokens) or fused via a prompt embedding adapter.
4. would grouping benefit from context awareness, not simply same call, same module name? Add a variant grouping by (module, k, cluster(q \oplus s)). If this outperforms (module, k) at the same keep-rate/compute, it indicates heterogeneity at fixed k was hurting.

---

> ### Author Response · Authors · 2025-12-03
>
> We thank the reviewer 5Lbj for taking the time to review this work. It is encouraging that the reviewer finds the implementation relatively easy and practical (no architectural changes required, integrates naturally with LoRA). We address the remaining concerns below.
>
> > Heterogeneous groups. Grouping by the k-th call mixes unlike contexts/prompts; apples-to-oranges grouping can inflate variance and misdirect updates.
>
> We would like to clarify that the method does try to address this concern in fact. To better illustrate our current approach to mmGRPO, we added Figure 2, which shows how module level groups are formed. In particular, we *only* group calls across the trajectories if they come from the same module of a given program. We further match the invocation call order to the best of our ability. We acknowledge that there will still be some deviations between the input prompts, even if a module invocation belongs to the same module, at the same step, on the same input (due to intermediate steps having distinct inputs/outputs).
>
> > Arbitrary partitioning risk. Module boundaries are designer choices; outcomes can depend on where you cut. The paper offers no merge/split robustness or guidance.
> > merge/split: what's the design criteria for choosing the boundary? also include ablation of merge/split of modules.
>
> Regarding module boundaries, we agree that program design is important, this is well documented in Shangyin et. al. 2025. This is a property of the LM program that’s inputted to mmGRPO, which is how many LLM-based systems are developed nowadays and we simply assume the program is given to the algorithm (we are not proposing any new programs) . To clarify these points, we have updated our paper with figures to better exemplify what LM programs are and how we form GRPO groups with mmGRPO.
>
> > Strong Markov-given-message assumption. It is assumed, not validated, that q fully summarizes upstream state; if serialization is lossy, gradients are biased.
>
> The writer of the LM program decides which inputs are passed to each module and it is often domain-specific. In principle, the programmer could choose to implement an LM program with a simple agent scaffold where the full history is passed to each module, where there would be no loss of the prior context.
>
> > specify the message q scheme. what information about all prefix goes into the message that by right is sufficient to encode everything?
> > following (2), curious to see a state-Incorporated module (learned latent / prompt embedding). Try a learned state appended to q: a small encoder s = f(upstream traces) whose vector is serialized (text or special tokens) or fused via a prompt embedding adapter.
>
> We included Figure 1 to show an example LM program invocation to illustrate the structure for q. In summary, the program logic decides which information is passed to each module and each program is written by a domain expert.
> We thank the reviewer for the grouping suggestion – we agree this fused approach is an interesting middle-ground between the declarative LM programs and monolithic setups where the full system is expressed as a single call to an LM. We think that this stateful variant is out-of-scope for our current work as we are focused on extending GRPO to existing LM programs.
>
> > would grouping benefit from context awareness, not simply same call, same module name? Add a variant grouping by (module, k, cluster(q \oplus s)). If this outperforms (module, k) at the same keep-rate/compute, it indicates heterogeneity at fixed k was hurting.
>
> We did not implement this variant in the current paper to keep the core contribution focused on “can GRPO be made to work for LM programs at all and whether it composes with prompt optimization?” rather than optimizing the grouping strategy itself. However, we agree that grouping by (module, call_order) vs. (module, semantic_similarity) could be an interesting direction for future work that focuses on developing other extensions of GRPO for multi-module systems: this would directly quantify how much within group context helps
>
> We thank the reviewer again for the thoughtful feedback and hope that these clarifications help highlight the contributions of this work more clearly.

---

### Official Review · Reviewer_ZN5E · 2025-10-30

**Soundness:** 3
**Presentation:** 2
**Contribution:** 2
**Rating:** 4
**Confidence:** 2

**Summary:**

This work introduces MMGRPO, a multi-module extension of  GRPO designed for modular AI systems that combine multiple language model calls and tools. MMGRPO groups calls by module and manages variable or interrupted trajectories. When combined with automatic prompt optimization, it improves accuracy over post-trained LMs and over prompt optimization alone. The optimizer is released as an open-source tool.

**Strengths:**

- The paper appears solid overall, providing a convincing answer to the research question it poses.

- Although it does not make a particularly strong claim, the motivation and reasoning are coherent.

- The proposed approach is conceptually simple, as it mainly applies GRPO to a multistep system rather than introducing a new or overly complex algorithm.

**Weaknesses:**

- The ablation study and experimental analysis, from a quantitative point of view, are limited. Reducing the empirical depth of the analysis.

- The evaluation relies exclusively on accuracy, which is inadequate for large language model assessment.
The authors should adopt standard LLM evaluation metrics, such as pass@k (and ideally maj@k, as used in DeepSeek), to provide a more comprehensive and reliable performance evaluation.

**Questions:**

Please address the weaknesses.

---

> ### Author Response · Authors · 2025-12-03
>
> We thank the reviewer for taking the time to review the paper. We are encouraged that the reviewer finds the paper solid overall, that the motivation and reasoning are coherent, and that the proposed extension is conceptually simple. We address the concerns inline below.
>
> > The ablation study and experimental analysis, from a quantitative point of view, are limited. Reducing the empirical depth of the analysis.
>
> Our key contribution is showing that combining  prompt optimization with weight optimization continues to show consistent gains even when the weight optimization is performed with an online RL method, in the context of compound AI systems. We demonstrate that prompt optimization (via the MIPRO prompt optimizer) and GRPO (as implemented with our mmGRPO extension) provide complementary benefits, and combined together, yield stronger results. We clarify that we describe mmGRPO as a necessary intermediate step towards extending policy-based online RL methods to multi-module compound AI systems (and it is the first such system).
>
> More specifically, Table 1 demonstrates three distinct mechanisms by which the BetterTogether strategy yields superior performance:
> 1. Overcoming Exploration Bottlenecks of the base policy: On complex multi-hop tasks, mmGRPO alone struggles if the initial policy is too weak to generate successful trajectories (HoVer: 60.2% vs. Baseline 59.5%). MIPRO effectively provides a "warm start" (63.4%), placing the model in a trainable region. The combination then jumps to 68.3%, showing that mmGRPO can exploit the better starting point to achieve significantly better results than either prompt optimization or reinforcement learning alone.
> 2. Orthogonality of Gains: In the privacy task, both methods achieve an individual score of 83.9%. However, their combination rises to 86.5%. This implies the methods learn complementary strategies solving different (sub)problems.
> 3. Performance Ceiling: While prompt optimization offers a strong baseline (Avg 70.0%), it consistently hits a performance ceiling when the model has not been updated parametrically. Adding mmGRPO reliably unlocks further gains (BetterTogether Avg 73.4%), confirming that weight updates can solve fine-grained optimization problems that are strictly inaccessible to prompt engineering
>
>
> > The evaluation relies exclusively on accuracy, which is inadequate for large language model assessment. The authors should adopt standard LLM evaluation metrics, such as pass@k (and ideally maj@k, as used in DeepSeek), to provide a more comprehensive and reliable performance evaluation.
>
> Our tasks require accuracy based evaluation as due to how they are structured (as shared below):
> 1. Each task is implemented as an LM program that produces a single final outcome per example: a class label for Banking77, a response for PAPILLON, or a verification verdict for HoVer.
> 2. At evaluation time, we run each program once per test example with a single sample, which is the case in the typical deployment of such pipelines.
> In our tasks, each program is intended to produce a single reliable output per input. Accuracy metric is also consistent with the prior work in this space [1].
>
> We thank the reviewer again for the thoughtful feedback. We posted a revised version of the paper with improved clarity and additional figures: Figure 1 shows an example invocation of a multi-hop retrieval LM program, and Figure 2 shows how mmGRPO makes groups. We hope that the above clarifications better situate both the empirical choices and the overall contribution.
>
> References
> [1] Yichen Jiang et al., [HoVer: A Dataset for Many-Hop Fact Extraction And Claim Verification](https://arxiv.org/abs/2011.03088), ACL 2020

---

### Official Review · Reviewer_qhaK · 2025-10-30

**Soundness:** 3
**Presentation:** 1
**Contribution:** 2
**Rating:** 2
**Confidence:** 3

**Summary:**

This paper proposed Multi-module GRPO, an extension to GRPO algorithm/implementation which allows grouping LM calls by module and weight optimization for each module separately. They motivate this through the need for optimizing compound AI systems that consist of multiple modules (e.g., deep research) and allow privacy-preserving applications. They also showed combining mmGRPO with prompt optimization techniques consistently improve performance across 3 tasks of single-stage intent-classification, two-stage privacy-conscious delegation program, and four-stage claim verification.

**Strengths:**

- The proposed method is addressing an increasingly relevant challenge in modern AI systems—optimizing multi-component or modular architectures rather than isolated models

**Weaknesses:**

- First, the paper is really poorly written, and suffers from significant clarity and presentation issues. It lacks a clear scientific structure and concrete running examples that could help readers understand the core problem and motivation. Introducing a concrete example early in the introduction would greatly improve readability. Certain terms get used for the first time without prior definition or explanation (see details in question section). Overall, the paper, in its current state, is not ready for publication and would benefit from substantial rewriting to improve presentation and clarity.

- The results section is also underdeveloped. The empirical evaluation is limited to a single table (Table 1) with no additional analysis, and ablation studies to justify the design choices or isolate the contributions of different components. The comparison baselines are also quite narrow.

- Finally, the advantages of mmGRPO over simpler methods such as prompt optimization, which require substantially less computational overhead, are not convincingly demonstrated (and are even acknowledged by the authors)

**Questions:**

1- [suggestion] The paper can benefit from a teaser/motivating figure that readily explains the idea/motivation behind the method.

2- [suggestion] A concrete example on LM program would have been helpful for the reader to understand what you mean by Control Flow, trajectory of module calls, etc.

3- [Line 191] certain terms get used for the first time without properly being introduced before. What does the author refer to student/teacher program In Section 3?

4- Do authors consider a separate module for the last step in each tasks? like label generation for classification, final response generation for PUPA etc.

5- why not including the trajectory-level GRPO as another baseline to better understand the contribution of grouping the trajectories (module-level)?

---

> ### Author Response · Authors · 2025-12-03
>
> We thank the reviewer for taking the time to review our paper and share their feedback. We are encouraged that the reviewer found our problem setting to be an increasingly relevant challenge in modern AI systems. We also thank the reviewer for their suggestions to improve the clarity of some concepts.
>
> We have now updated the paper with a new running example, introduced early in the paper and re-used throughout the methods section. The goal of this running example is to better illustrate (a) control flow and trajectory in a compound AI system (b) how (disjoint) module inputs are constructed, and (c) how module-level groups are formed in GRPO. We have also included Figure 1, showing an example invocation of a multi-hop retrieval LM program, and Figure 2 showing how mmGRPO makes groups. We clarify the reviewer’s concerns inline below.
>
>
> > advantages of mmGRPO over … prompt optimization, which require substantially less computational overhead, are not convincingly demonstrated (and are even acknowledged by the authors)
>
> Our key contribution is showing that combining  prompt optimization with weight optimization continues to show consistent gains even when the weight optimization is performed with an online RL method, in the context of compound AI systems. We demonstrate that prompt optimization (via the MIPRO prompt optimizer) and GRPO (as implemented with our mmGRPO extension) provide complementary benefits, and combined together, yield stronger results. We clarify that we describe mmGRPO as a necessary intermediate step towards extending policy-based online RL methods to multi-module compound AI systems (and it is the first such system).
>
> > no additional analysis
>
> We appreciate your comment on this and we apologize for the confusion. Table 1 demonstrates three distinct mechanisms by which the BetterTogether strategy yields superior performance:
> 1. Overcoming Exploration Bottlenecks of the base policy: On complex multi-hop tasks, mmGRPO alone struggles if the initial policy is too weak to generate successful trajectories (HoVer: 60.2% vs. Baseline 59.5%). MIPRO effectively provides a "warm start" (63.4%), placing the model in a trainable region. The combination then jumps to 68.3%, showing that mmGRPO can exploit the better starting point to achieve significantly better results than either prompt optimization or reinforcement learning alone.
> 2. Orthogonality of Gains: In the privacy task, both methods achieve an individual score of 83.9%. However, their combination rises to 86.5%. This implies the methods learn complementary strategies solving different (sub)problems.
> 3. Performance Ceiling: While prompt optimization offers a strong baseline (Avg 70.0%), it consistently hits a performance ceiling when the model has not been updated parametrically. Adding mmGRPO reliably unlocks further gains (BetterTogether Avg 73.4%), confirming that weight updates can solve fine-grained optimization problems that are strictly inaccessible to prompt engineering.
> We have updated the experimental results section to better emphasize the contributions mentioned above.
>
> > Do authors consider a separate module for the last step in each task? like label generation for classification, final response generation for PUPA etc.
>
> We discuss details of the compound AI systems used in our experiments in section 5.1 with the code snippets for the LM programs themselves being in Section C of the  For classification, a chain of thought module is used to predict the classification label. For PUPA, we use a 2-module system obtained from [1], whose last module rewrites the untrusted LLMs response with the trusted LLM (as hypothesized by the reviewer), and for HoVer, we create a 4-hop system, which performs query-writing, retrieval and summarization for 4 hops, returning the set of all documents.
>
>
> > 2- [suggestion] A concrete example on LM program would have been helpful for the reader to understand what you mean by Control Flow, trajectory of module calls, etc.
>
> We have added a figure showing an example of an LM program invocation for multi-hop retrieval. We also have code snippets for all of our LM programs in Section C of the appendix.
>
>
> > 3- [Line 191] certain terms get used for the first time without properly being introduced before. What does the author refer to student/teacher program In Section 3?
>
> We have removed the paragraph which discusses the possibility of using a teacher/student training setup as we do not report any experiments with this approach, and instead we leave it for future work.
>
>
> **References**
>
> [1] Siyan Li et al., [PAPILLON: Privacy Preservation from Internet-based and Local Language Model Ensembles](https://aclanthology.org/2025.naacl-long.173/), ACL 2025

---

### Author Response · Authors · 2025-12-03

# Summary of overall contributions
We thank the reviewers for their detailed feedback. The main goal of this work  is to study **whether prompt optimization and online RL compose in compound AI systems**. We show that combining prompt optimization (MIPROv2) with GRPO (via our mmGRPO extension) yields *consistent, complementary gains* even when the weight optimization is an online RL method. To apply online RL to arbitrary and multi-module LM programs, we define the mmGRPO strategy, which is  to our knowledge, the first online RL implementation tested on compound AI systems of even mild complexity (e.g., with two or more modules). Many comments (especially from qhaK and LPF6) focused on mmGRPO itself and on whether it beats prompt optimization., but missed its primary role in enabling extending the BetterTogether result to GRPO-style algorithms, not to claim that mmGRPO alone strictly dominates strong prompt baselines or is the strongest possible RL approach. Thanks to the reviewers’ feedback, we have improved the writing to further clarify this claim, and we highlight in the text that BetterTogether consistently outperforms both MIPROv2 and mmGRPO across all three LM programs.

# Clarity and framing
A large fraction of the critical feedback (qhaK, 5Lbj, LPF6) stemmed from misunderstandings about the **LM program setup, and what is assumed vs. optimized**. In response, we substantially revised the presentation. The new version adds:

* **A running LM-program example** that is referenced across methods and experiments.
* **Figure 1**, showing an example multi-hop retrieval LM program with explicit control flow and module inputs/outputs.
* **Figure 2**, illustrating how mmGRPO forms module-level GRPO groups.
* **Appendix Section D**, showing a full example trace of a LM program with disjoint modules

We also clarify the scope more directly: our method assumes that **an LM program for a task is given** and our goal is to optimize the prompts and weights of this already designed program (addressing comments from qhaK, 5Lbj, LPF6). The programmer decides which information is passed into each module and where module boundaries lie; we do not modify or learn the program structure itself. This matches many state-of-the-art compound AI systems, and Shangyin et al. 2025 document that such LM programs often outperform agentic baselines on their respective tasks. We now explicitly say that the paper is not proposing new LM programs or automatically discovering boundaries; it operates directly in this LM-program regime.

# Experimental scope
Some reviewers commented on limited ablations. We clarify that our empirical goal is to test **BetterTogether and mmGRPO on three LM programs** (Banking77, PAPILLON, HoVer), each with a strong task-specific LM-program baseline and MIPROv2 as a prompt-optimization baseline, rather than exhaustively ablating every mmGRPO design choice. Because mmGRPO is, to our knowledge, the first GRPO-style online RL method that directly handles multi-module LM programs, there is no existing online RL baseline in this setting. In the single-module Banking77 case, mmGRPO reduces to standard GRPO grouping, and BetterTogether still improves over both GRPO and MIPROv2, which isolates the benefit in a setting where a vanilla GRPO baseline is available.

Overall, most of the critical points were about **framing and clarity**, not about the core idea itself. We believe the revised version addresses these misunderstandings directly. LM programs are taken as given; mmGRPO is the first GRPO style online RL method that works in this regime; and the main empirical takeaway is that **BetterTogether reliably improves over both prompt optimization and online RL alone in compound AI systems**.

---

### Meta-Review · Area_Chair_k4sm · 2026-01-09

**Summary:**

This paper studies how to extend Group Relative Policy Optimization (GRPO) to modular and compound AI systems that interleave multiple language model calls with distinct prompts and external tools. The authors introduce mmGRPO, a simple multi-module generalization that groups LM calls by module across rollouts and accommodates variable-length and interrupted trajectories. Empirically, when combined with automatic prompt optimization, mmGRPO is claimed to yield consistent performance gains across a range of tasks, including classification, multi-hop search, and privacy-preserving delegation, and is released as an open-source optimization framework for compound AI systems.

**Reviewer Concerns:**

- Presentation and clarity issues: The paper suffers from significant clarity and organizational problems, including undefined terminology, a lack of concrete running examples, and insufficient explanation of key concepts.

- Underdeveloped empirical evaluation: The experimental section is limited in depth, relying on a small number of metrics (primarily accuracy), a narrow set of baselines, and minimal analysis. There is a lack of ablations to justify design choices, isolate the contribution of module-level grouping, or study robustness to alternative design decisions.

- Unclear advantage over simpler baselines: The benefits of mmGRPO over substantially simpler and cheaper alternatives—most notably prompt optimization alone—are not convincingly demonstrated. In some settings, mmGRPO does not outperform prompt optimization, raising questions about its cost–benefit tradeoff.

- Sensitivity to module partitioning and grouping strategy: The effectiveness of grouping by module and call index may depend heavily on arbitrary design choices. There is no analysis of merge/split robustness, grouping heterogeneity, or whether more context-aware grouping strategies would yield better or more stable learning.

- Missing or insufficiently discussed limitations: Important limitations—such as the Markovian assumption, computational overhead, and scalability—are either not explicitly acknowledged or not discussed in the limitations section, despite being central to the method’s applicability.

**Reviewer Scores:**

The overall consensus is towards rejection. I do not believe any reviewer will strongly advocate for acceptance due to the flagged limitations.

---

### Decision · Program_Chairs · 2026-01-26

Reject